# PIXART-$\alpha$: FAST TRAINING OF DIFFUSION TRANS-FORMER FOR PHOTOREALISTIC TEXT-TO-IMAGE SYNTHESIS

**Junsong Chen**[1,2,3*], **Jincheng Yu**[1,4*], **Chongjian Ge**[1,3*], **Lewei Yao**[1,4*], **Enze Xie**[1†],
**Yue Wu**[1], **Zhongdao Wang**[1], **James Kwok**[4], **Ping Luo**[3], **Huchuan Lu**[2], **Zhenguo Li**[1]

[1]Huawei Noah's Ark Lab    [2]Dalian University of Technology    [3]HKU    [4]HKUST

`jschen@mail.dlut.edu.cn, rhettgee@connect.hku.hk,`
`{yujincheng4,yao.lewei,xie.enze,Li.Zhenguo}@huawei.com`

Project Page: https://pixart-alpha.github.io/

## ABSTRACT

The most advanced text-to-image (T2I) models require significant training costs (*e.g.*, millions of GPU hours), seriously hindering the fundamental innovation for the AIGC community while increasing $CO_2$ emissions. This paper introduces PIXART-$\alpha$, a Transformer-based T2I diffusion model whose image generation quality is competitive with state-of-the-art image generators (*e.g.*, Imagen, SDXL, and even Midjourney), reaching near-commercial application standards. Additionally, it supports high-resolution image synthesis up to 1024 × 1024 resolution with low training cost, as shown in Figure 1 and 2. To achieve this goal, three core designs are proposed: (1) Training strategy decomposition: We devise three distinct training steps that respectively optimize pixel dependency, text-image alignment, and image aesthetic quality; (2) Efficient T2I Transformer: We incorporate cross-attention modules into Diffusion Transformer (DiT) to inject text conditions and streamline the computation-intensive class-condition branch; (3) High-informative data: We emphasize the significance of concept density in text-image pairs and leverage a large Vision-Language model to auto-label dense pseudo-captions to assist text-image alignment learning. As a result, PIXART-$\alpha$'s training speed markedly surpasses existing large-scale T2I models, *e.g.*, PIXART-$\alpha$ only takes 12% of Stable Diffusion v1.5's training time ($\sim$753 *vs.* $\sim$6,250 A100 GPU days), saving nearly $300,000 ($28,400 *vs.* $320,000) and reducing 90% $CO_2$ emissions. Moreover, compared with a larger SOTA model, RAPHAEL, our training cost is merely 1%. Extensive experiments demonstrate that PIXART-$\alpha$ excels in image quality, artistry, and semantic control. We hope PIXART-$\alpha$ will provide new insights to the AIGC community and startups to accelerate building their own high-quality yet low-cost generative models from scratch.

## 1 INTRODUCTION

Recently, the advancement of text-to-image (T2I) generative models, such as DALL·E 2 (OpenAI, 2023), Imagen (Saharia et al., 2022), and Stable Diffusion (Rombach et al., 2022) has started a new era of photorealistic image synthesis, profoundly impacting numerous downstream applications, such as image editing (Kim et al., 2022), video generation (Wu et al., 2022), 3D assets creation (Poole et al., 2022), *etc.*

However, the training of these advanced models demands immense computational resources. For instance, training SDv1.5 (Podell et al., 2023) necessitates 6K A100 GPU days, approximately costing $320,000, and the recent larger model, RAPHAEL (Xue et al., 2023b), even costs 60K A100 GPU days – requiring around $3,080,000, as detailed in Table 2. Additionally, the training contributes substantial $CO_2$ emissions, posing environmental stress; *e.g.* RAPHAEL's (Xue et al., 2023b) train-

---

*Equal contribution. Work done during the internships of the four students at Huawei Noah's Ark Lab.
† Project lead and corresponding author.

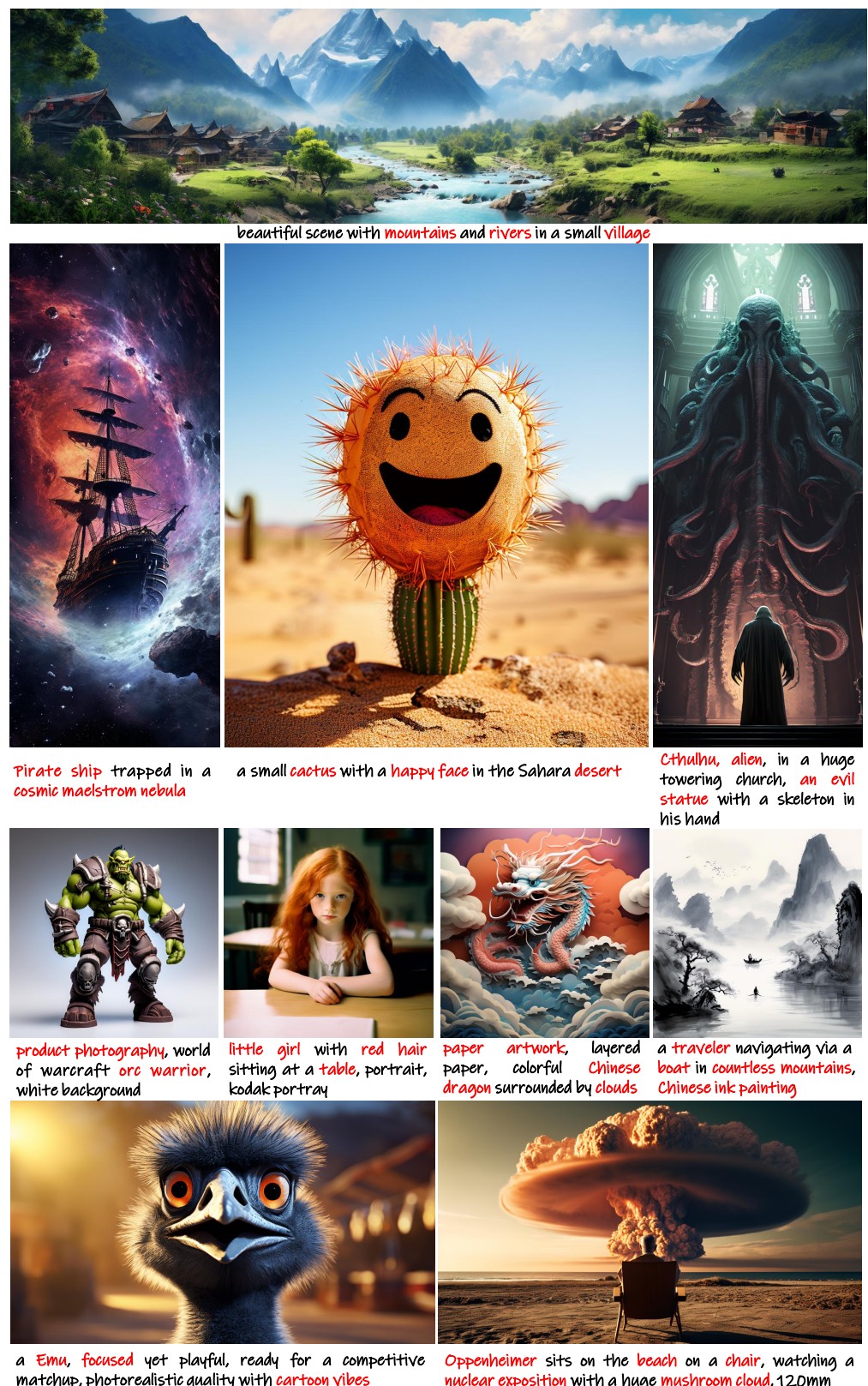

Figure 1: Samples produced by PIXART-α exhibit exceptional quality, characterized by a remarkable level of fidelity and precision in adhering to the provided textual descriptions.

ing results in 35 tons of $CO_2$ emissions, equivalent to the amount one person emits over 7 years, as shown in Figure 2. Such a huge cost imposes significant barriers for both the research community and entrepreneurs in accessing those models, causing a significant hindrance to the crucial advancement of the AIGC community. Given these challenges, a pivotal question arises: *Can we develop a high-quality image generator with affordable resource consumption?*

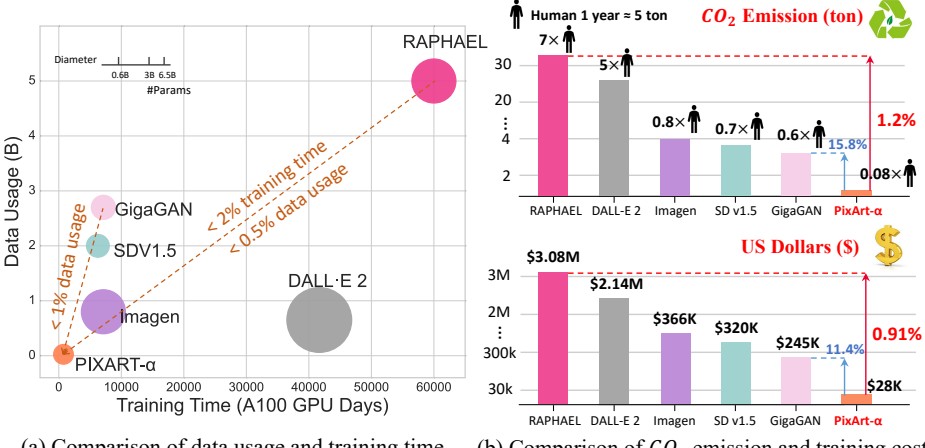

(a) Comparison of data usage and training time  (b) Comparison of $CO_2$ emission and training cost

Figure 2: Comparisons of $CO_2$ emissions[1] and training cost[2] among T2I generators. PIXART-$\alpha$ achieves an exceptionally low training cost of \$28,400. Compared to RAPHAEL, our $CO_2$ emissions and training costs are merely 1.2% and 0.91%, respectively.

In this paper, we introduce PIXART-$\alpha$, which significantly reduces computational demands of training while maintaining competitive image generation quality to the current state-of-the-art image generators, as illustrated in Figure 1. To achieve this, we propose three core designs:

**Training strategy decomposition.** We decompose the intricate text-to-image generation task into three streamlined subtasks: (1) learning the pixel distribution of natural images, (2) learning text-image alignment, and (3) enhancing the aesthetic quality of images. For the first subtask, we propose initializing the T2I model with a low-cost class-condition model, significantly reducing the learning cost. For the second and third subtasks, we formulate a training paradigm consisting of pretraining and fine-tuning: pretraining on text-image pair data rich in information density, followed by fine-tuning on data with superior aesthetic quality, boosting the training efficiency.

**Efficient T2I Transformer.** Based on the Diffusion Transformer (DiT) (Peebles & Xie, 2023), we incorporate cross-attention modules to inject text conditions and streamline the computation-intensive class-condition branch to improve efficiency. Furthermore, we introduce a re-parameterization technique that allows the adjusted text-to-image model to load the original class-condition model's parameters directly. Consequently, we can leverage prior knowledge learned from ImageNet (Deng et al., 2009) about natural image distribution to give a reasonable initialization for the T2I Transformer and accelerate its training.

**High-informative data.** Our investigation reveals notable shortcomings in existing text-image pair datasets, exemplified by LAION (Schuhmann et al., 2021), where textual captions often suffer from a lack of informative content (*i.e.*, typically describing only a partial of objects in the images) and a severe long-tail effect (*i.e.*, with a large number of nouns appearing with extremely low frequencies). These deficiencies significantly hamper the training efficiency for T2I models and lead to millions of iterations to learn stable text-image alignments. To address them, we propose an auto-labeling pipeline utilizing the state-of-the-art vision-language model (LLaVA (Liu et al., 2023)) to generate captions on the SAM (Kirillov et al., 2023). Referencing in Section 2.4, the SAM dataset is advantageous due to its rich and diverse collection of objects, making it an ideal resource for creating high-information-density text-image pairs, more suitable for text-image alignment learning.

---

[1]The method for estimating $CO_2$ emissions follows Alexandra Sasha Luccioni (2022).
[2]The training cost refers to the cloud GPU pricing from Microsoft (2023) Azure in September 20, 2023.

Our effective designs result in remarkable training efficiency for our model, costing only 753 A100 GPU days and $28,400. As demonstrated in Figure 2, our method consumes less than 1.25% training data volume compared to SDv1.5 and costs less than 2% training time compared to RAPHAEL. Compared to RAPHAEL, our training costs are only 1%, saving approximately $3,000,000 (PIXART-$\alpha$'s $28,400 *vs*. RAPHAEL's $3,080,000). Regarding generation quality, our user study experiments indicate that PIXART-$\alpha$ offers superior image quality and semantic alignment compared to existing SOTA T2I models (*e.g*., DALL·E 2 (OpenAI, 2023), Stable Diffusion (Rombach et al., 2022), *etc*.), and its performance on T2I-CompBench (Huang et al., 2023) also evidences our advantage in semantic control. We hope our attempts to train T2I models efficiently can offer valuable insights for the AIGC community and help more individual researchers or startups create their own high-quality T2I models at lower costs.

## 2 METHOD

### 2.1 MOTIVATION

The reasons for slow T2I training lie in two aspects: the training pipeline and the data.

The T2I generation task can be decomposed into three aspects: **Capturing Pixel Dependency:** Generating realistic images involves understanding intricate pixel-level dependencies within images and capturing their distribution; **Alignment between Text and Image:** Precise alignment learning is required for understanding how to generate images that accurately match the text description; **High Aesthetic Quality:** Besides faithful textual descriptions, being aesthetically pleasing is another vital attribute of generated images. Current methods entangle these three problems together and directly train from scratch using vast amount of data, resulting in inefficient training. To solve this issue, we disentangle these aspects into three stages, as will be described in Section 2.2.

Another problem, depicted in Figure 3, is with the quality of captions of the current dataset. The current text-image pairs often suffer from text-image misalignment, deficient descriptions, infrequent diverse vocabulary usage, and inclusion of low-quality data. These problems introduce difficulties in training, resulting in unnecessarily millions of iterations to achieve stable alignment between text and images. To address this challenge, we introduce an innovative auto-labeling pipeline to generate precise image captions, as will be described in Section 2.4.

| Problems | Text-image misalignment | Deficient descriptions | Infrequent vocabulary |
|---|---|---|---|
| **Samples** |  |  |  |
| **Raw caption** | What science says about pu'erh tea? | AH1370/1950 Saudi Arabia Gold One Guinea MS-63 NGC | 2018 Kawasaki Jet Ski Ultra 310LX in Unionville, Virginia |
| **LLaVA refined caption** | The image features a close-up of a cup of tea with a saucer on a wooden table. The tea is described as "pu'erh tea," which is a type of Chinese tea known for its health benefits. The scene is set in a dimly lit room. The presence of a potted plant in the background adds a touch of nature and freshness to the scene. | The image shows a man working on scuba diving equipment at Blue Water Divers. The man is sitting at a table, working on a piece of equipment, possibly fixing or adjusting it. The scene is set in a workshop or a store, with various tools and equipment visible in the background. | The image features a man riding a jet ski on a body of water. The jet ski is green and white, and it is being used for recreational purposes. The man is smiling, indicating that he is enjoying his time on the water. The scene is set in a beach area. |

Figure 3: LAION raw captions *v.s* LLaVA refined captions. LLaVA provides high-information-density captions that aid the model in grasping more concepts per iteration and boost text-image alignment efficiency.

### 2.2 TRAINING STRATEGY DECOMPOSITION

The model's generative capabilities can be gradually optimized by partitioning the training into three stages with different data types.

**Stage1: Pixel dependency learning.** The current class-guided approach (Peebles & Xie, 2023) has shown exemplary performance in generating semantically coherent and reasonable pixels in individual images. Training a class-conditional image generation model (Peebles & Xie, 2023) for natural images is relatively easy and inexpensive, as explained in Appendix A.5. Additionally, we

find that a suitable initialization can significantly boost training efficiency. Therefore, we boost our model from an ImageNet-pretrained model, and the architecture of our model is designed to be compatible with the pretrained weights.

**Stage2: Text-image alignment learning.** The primary challenge in transitioning from pretrained class-guided image generation to text-to-image generation is on how to achieve accurate alignment between significantly increased text concepts and images.

This alignment process is not only time-consuming but also inherently challenging. To efficiently facilitate this process, we construct a dataset consisting of precise text-image pairs with high concept density. The data creation pipeline will be described in Section 2.4. By employing accurate and information-rich data, our training process can efficiently handle a larger number of nouns in each iteration while encountering considerably less ambiguity compared to previous datasets. This strategic approach empowers our network to align textual descriptions with images effectively.

**Stage3: High-resolution and aesthetic image generation.** In the third stage, we fine-tune our model using high-quality aesthetic data for high-resolution image generation. Remarkably, we observe that the adaptation process in this stage converges significantly faster, primarily owing to the strong prior knowledge established in the preceding stages.

Decoupling the training process into different stages significantly alleviates the training difficulties and achieves highly efficient training.

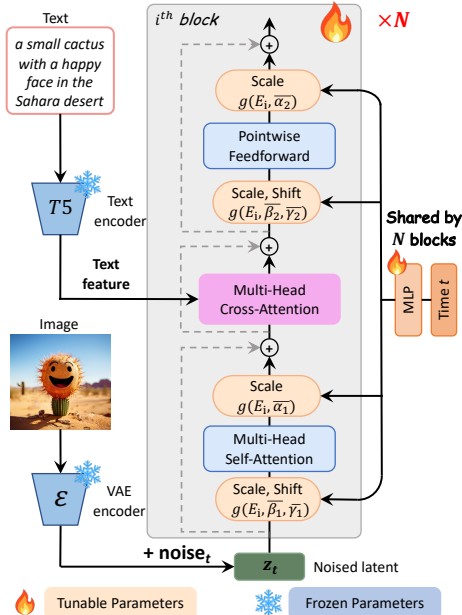

Figure 4: Model architecture of PIXART-$\alpha$. A cross-attention module is integrated into each block to inject textual conditions. To optimize efficiency, all blocks share the same adaLN-single parameters for time conditions.

## 2.3 EFFICIENT T2I TRANSFORMER

PIXART-$\alpha$ adopts the Diffusion Transformer (DiT) (Peebles & Xie, 2023) as the base architecture and innovatively tailors the Transformer blocks to handle the unique challenges of T2I tasks, as depicted in Figure 4. Several dedicated designs are proposed as follows:

- *Cross-Attention layer.* We incorporate a multi-head cross-attention layer to the DiT block. It is positioned between the self-attention layer and feed-forward layer so that the model can flexibly interact with the text embedding extracted from the language model. To facilitate the pretrained weights, we initialize the output projection layer in the cross-attention layer to zero, effectively acting as an identity mapping and preserving the input for the subsequent layers.

- *AdaLN-single.* We find that the linear projections in the adaptive normalization layers (Perez et al., 2018) (*adaLN*) module of the DiT account for a substantial proportion (27%) of the parameters. Such a large number of parameters is not useful since the class condition is not employed for our T2I model. Thus, we propose adaLN-single, which only uses time embedding as input in the first block for independent control (shown on the right side of Figure 4). Specifically, in the $i$th block, let $S^{(i)} = [\beta_1^{(i)}, \beta_2^{(i)}, \gamma_1^{(i)}, \gamma_2^{(i)}, \alpha_1^{(i)}, \alpha_2^{(i)}]$ be a tuple of all the scales and shift parameters in *adaLN*. In the DiT, $S^{(i)}$ is obtained through a block-specific MLP $S^{(i)} = f^{(i)}(c+t)$, where $c$ and $t$ denotes the class condition and time embedding, respectively. However, in adaLN-single, one global set of shifts and scales are computed as $\overline{S} = f(t)$ only at the first block which is shared across all the blocks. Then, $S^{(i)}$ is obtained as $S^{(i)} = g(\overline{S}, E^{(i)})$, where $g$ is a summation function, and $E^{(i)}$ is a layer-specific trainable embedding with the same shape as $\overline{S}$, which adaptively adjusts the scale and shift parameters in different blocks.

- *Re-parameterization.* To utilize the aforementioned pretrained weights, all $E^{(i)}$'s are initialized to values that yield the same $S^{(i)}$ as the DiT without $c$ for a selected $t$ (empirically, we use $t = 500$). This design effectively replaces the layer-specific MLPs with a global MLP and layer-specific trainable embeddings while preserving compatibility with the pretrained weights.

Experiments demonstrate that incorporating a global MLP and layer-wise embeddings for time-step information, as well as cross-attention layers for handling textual information, persists the model's generative abilities while effectively reducing its size.

## 2.4 DATASET CONSTRUCTION

**Image-text pair auto-labeling.** The captions of the LAION dataset exhibit various issues, such as text-image misalignment, deficient descriptions, and infrequent vocabulary as shown in Figure 3. To generate captions with high information density, we leverage the state-of-the-art vision-language model LLaVA (Liu et al., 2023). Employing the prompt, "*Describe this image and its style in a very detailed manner*", we have significantly improved the quality of captions, as shown in Figure 3.

However, it is worth noting that the LAION dataset predominantly comprises of simplistic product previews from shopping websites, which are not ideal for training text-to-image generation that seeks diversity in object combinations. Consequently, we have opted to utilize the SAM dataset (Kirillov et al., 2023), which is originally used for segmentation tasks but features imagery rich in diverse objects. By applying LLaVA to SAM, we have successfully acquired high-quality text-image pairs characterized by a high concept density, as shown in Figure 10 and Figure 11 in the Appendix.

In the third stage, we construct our training dataset by incorporating JourneyDB (Pan et al., 2023) and a 10M internal dataset to enhance the aesthetic quality of generated images beyond realistic photographs. Refer to Appendix A.5 for details.

As a result, we show the vocabulary analysis (NLTK, 2023) in Table 1, and we define the valid distinct nouns as those appearing more than 10 times in the dataset. We apply LLaVA on LAION to generate LAION-LLaVA. The LAION dataset has 2.46 M distinct nouns, but only 8.5% are valid. This valid noun proportion significantly increases from 8.5% to 13.3% with LLaVA-labeled captions. Despite

Table 1: Statistics of noun concepts for different datasets. **VN**: valid distinct nouns (appearing more than 10 times); **DN**: total distinct nouns; **Average**: average noun count per image.

| Dataset | VN/DN | Total Noun | Average |
|---------|-------|------------|---------|
| LAION | 210K/2461K = 8.5% | 72.0M | 6.4/Img |
| LAION-LLaVA | 85K/646K = 13.3% | 233.9M | 20.9/Img |
| SAM-LLaVA | 23K/124K = 18.6% | 327.9M | 29.3/Img |
| Internal | 152K/582K = 26.1% | 136.6M | 12.2/Img |

LAION's original captions containing a staggering 210K distinct nouns, its total noun number is a mere 72M. However, LAION-LLaVA contains 234M noun numbers with 85K distinct nouns, and the average number of nouns per image increases from 6.4 to 21, indicating the incompleteness of the original LAION captions. Additionally, SAM-LLaVA outperforms LAION-LLaVA with a total noun number of 328M and 30 nouns per image, demonstrating SAM contains richer objectives and superior informative density per image. Lastly, the internal data also ensures sufficient valid nouns and average information density for fine-tuning. LLaVA-labeled captions significantly increase the valid ratio and average noun count per image, improving concept density.

## 3 EXPERIMENT

This section begins by outlining the detailed training and evaluation protocols. Subsequently, we provide comprehensive comparisons across three main metrics. We then delve into the critical designs implemented in PIXART-$\alpha$ to achieve superior efficiency and effectiveness through ablation studies. Finally, we demonstrate the versatility of our PIXART-$\alpha$ through application extensions.

## 3.1 IMPLEMENTATION DETAILS

**Training Details.** We follow Imagen (Saharia et al., 2022) and DeepFloyd (DeepFloyd, 2023) to employ the T5 large language model (*i.e.*, 4.3B Flan-T5-XXL) as the text encoder for conditional

Table 2: We thoroughly compare the PIXART-$\alpha$ with recent T2I models, considering several essential factors: model size, the total volume of training images, COCO FID-30K scores (zero-shot), and the computational cost (GPU days[3]). Our highly effective approach significantly reduces resource consumption, including training data usage and training time. The baseline data is sourced from GigaGAN (Kang et al., 2023). '+' in the table denotes an unknown internal dataset size.

| Method | Type | #Params | #Images | FID-30K↓ | GPU days |
|--------|------|---------|---------|----------|----------|
| DALL·E | Diff | 12.0B | 250M | 27.50 | - |
| GLIDE | Diff | 5.0B | 250M | 12.24 | - |
| LDM | Diff | 1.4B | 400M | 12.64 | - |
| DALL·E 2 | Diff | 6.5B | 650M | 10.39 | 41,667 A100 |
| SDv1.5 | Diff | 0.9B | 2000M | 9.62 | 6,250 A100 |
| GigaGAN | GAN | 0.9B | 2700M | 9.09 | 4,783 A100 |
| Imagen | Diff | 3.0B | 860M | 7.27 | 7,132 A100 |
| RAPHAEL | Diff | 3.0B | 5000M+ | 6.61 | 60,000 A100 |
| PIXART-$\alpha$ | Diff | 0.6B | 25M | 7.32 | 753 A100 |

feature extraction, and use DiT-XL/2 (Peebles & Xie, 2023) as our base network architecture. Unlike previous works that extract a standard and fixed 77 text tokens, we adjust the length of extracted text tokens to 120, as the caption curated in PIXART-$\alpha$ is much denser to provide more fine-grained details. To capture the latent features of input images, we employ a pre-trained and frozen VAE from LDM (Rombach et al., 2022). Before feeding the images into the VAE, we resize and center-crop them to have the same size. We also employ multi-aspect augmentation introduced in SDXL (Podell et al., 2023) to enable arbitrary aspect image generation. The AdamW optimizer (Loshchilov & Hutter, 2017) is utilized with a weight decay of 0.03 and a constant 2e-5 learning rate. Our final model is trained on 64 V100 for approximately 26 days. See more details in Appendix A.5.

**Evaluation Metrics.** We comprehensively evaluate PIXART-$\alpha$ via three primary metrics, *i.e.*, Fréchet Inception Distance (FID) (Heusel et al., 2017) on MSCOCO dataset (Lin et al., 2014), compositionality on T2I-CompBench (Huang et al., 2023), and human-preference rate on user study.

## 3.2 PERFORMANCE COMPARISONS AND ANALYSIS

**Fidelity Assessment.** The FID is a metric to evaluate the quality of generated images. The comparison between our method and other methods in terms of FID and their training time is summarized in Table 2. When tested for zero-shot performance on the COCO dataset, PIXART-$\alpha$ achieves a FID score of 7.32. It is particularly notable as it is accomplished in merely 12% of the training time (753 *vs.* 6250 A100 GPU days) and merely 1.25% of the training samples (25M *vs.* 2B images) relative to the second most efficient method. Compared to state-of-the-art methods typically trained using substantial resources, PIXART-$\alpha$ remarkably consumes approximately 2% of the training resources while achieving a comparable FID performance. Although the best-performing model (RAPHEAL) exhibits a lower FID, it relies on unaffordable resources (*i.e.*, $200\times$ more training samples, $80\times$ longer training time, and $5\times$ more network parameters than PIXART-$\alpha$). We argue that FID may not be an appropriate metric for image quality evaluation, and it is more appropriate to use the evaluation of human users, as stated in Appendix A.8. We leave scaling of PIXART-$\alpha$ for future exploration for performance enhancement.

**Alignment Assessment.** Beyond the above evaluation, we also assess the alignment between the generated images and text condition using T2I-Compbench (Huang et al., 2023), a comprehensive benchmark for evaluating the compositional text-to-image generation capability. As depicted in Table 3, we evaluate several crucial aspects, including attribute binding, object relationships, and complex compositions. PIXART-$\alpha$ exhibited outstanding performance across nearly all (5/6) evaluation metrics. This remarkable performance is primarily attributed to the text-image alignment learning in Stage 2 training described in Section 2.2, where high-quality text-image pairs were leveraged to achieve superior alignment capabilities.

---

[3]To ensure fairness, we convert the V100 GPU days (1656) of our training to A100 GPU days (753), assuming a $2.2\times$ speedup in U-Net training on A100 compared to V100, or equivalent to 332 A100 GPU days with a $5\times$ speedup in Transformer training, as per Rombach et al. (2022); NVIDIA (2023).

Table 3: Alignment evaluation on T2I-CompBench. PIXART-$\alpha$ demonstrated exceptional performance in attribute binding, object relationships, and complex compositions, indicating our method achieves superior compositional generation ability. We highlight the best value in blue, and the second-best value in green. The baseline data are sourced from Huang et al. (2023).

| Model | Attribute Binding | | | Object Relationship | | Complex↑ |
|---|---|---|---|---|---|---|
| | Color ↑ | Shape↑ | Texture↑ | Spatial↑ | Non-Spatial↑ | |
| Stable v1.4 | 0.3765 | 0.3576 | 0.4156 | 0.1246 | 0.3079 | 0.3080 |
| Stable v2 | 0.5065 | 0.4221 | 0.4922 | 0.1342 | 0.3096 | 0.3386 |
| Composable v2 | 0.4063 | 0.3299 | 0.3645 | 0.0800 | 0.2980 | 0.2898 |
| Structured v2 | 0.4990 | 0.4218 | 0.4900 | 0.1386 | 0.3111 | 0.3355 |
| Attn-Exct v2 | 0.6400 | 0.4517 | 0.5963 | 0.1455 | 0.3109 | 0.3401 |
| GORS | 0.6603 | 0.4785 | 0.6287 | 0.1815 | 0.3193 | 0.3328 |
| Dalle-2 | 0.5267 | 0.4747 | 0.5804 | 0.1283 | 0.3078 | 0.2967 |
| SDXL | 0.5879 | 0.4687 | 0.5299 | 0.2133 | 0.3119 | 0.3237 |
| PIXART-$\alpha$ | 0.6690 | 0.4927 | 0.6477 | 0.2064 | 0.3197 | 0.3433 |

**User Study.** While quantitative evaluation metrics measure the overall distribution of two image sets, they may not comprehensively evaluate the visual quality of the images. Consequently, we conducted a user study to supplement our evaluation and provide a more intuitive assessment of PIXART-$\alpha$'s performance. Since user study involves human evaluators and can be time-consuming, we selected the top-performing models, namely DALLE-2, SDv2, SDXL, and DeepFloyd, which are accessible through APIs and capable of generating images.

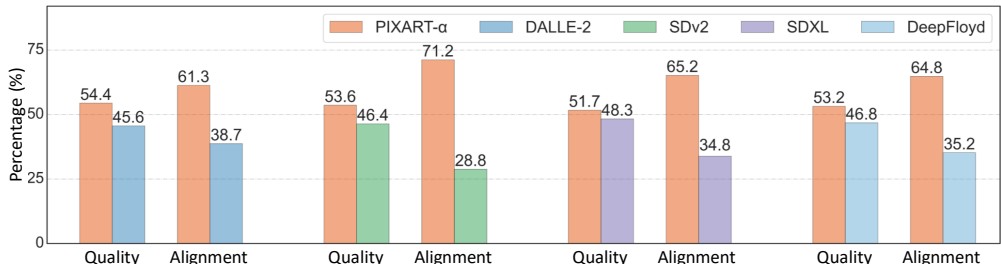

Figure 5: User study on 300 fixed prompts from Feng et al. (2023). The ratio values indicate the percentages of participants preferring the corresponding model. PIXART-$\alpha$ achieves a superior performance in both quality and alignment.

For each model, we employ a consistent set of 300 prompts from Feng et al. (2023) to generate images. These images are then distributed among 50 individuals for evaluation. Participants are asked to rank each model based on the perceptual quality of the generated images and the precision of alignments between the text prompts and the corresponding images. The results presented in Figure 5 clearly indicate that PIXART-$\alpha$ excels in both higher fidelity and superior alignment. For example, compared to SDv2, a current top-tier T2I model, PIXART-$\alpha$ exhibits a 7.2% improvement in image quality and a substantial 42.4% enhancement in alignment.

## 3.3 ABLATION STUDY

We then conduct ablation studies on the crucial modifications discussed in Section 2.3, including structure modifications and re-parameterization design. In Figure 6, we provide visual results and perform a FID analysis. We randomly choose 8 prompts from the SAM test set for visualization and compute the zero-shot FID-5K score on the SAM dataset. Details are described below.

"*w/o re-param*" results are generated from the model trained from scratch without re-parameterization design. We supplemented with an additional 200K iterations to compensate for the missing iterations from the pretraining stage for a fair comparison. "*adaLN*" results are from the model following the DiT structure to use the sum of time and text feature as input to the MLP layer for the scale and shift parameters within each block. "*adaLN-single*" results are obtained from the model using Transformer blocks with the adaLN-single module in Section 2.3. In both "*adaLN*" and "*adaLN-single*", we employ the re-parameterization design and training for 200K iterations.

As depicted in Figure 6, despite "*adaLN*" performing lower FID, its visual results are on par with our "*adaLN-single*" design. The GPU memory consumption of "*adaLN*" is 29GB, whereas "*adaLN-*

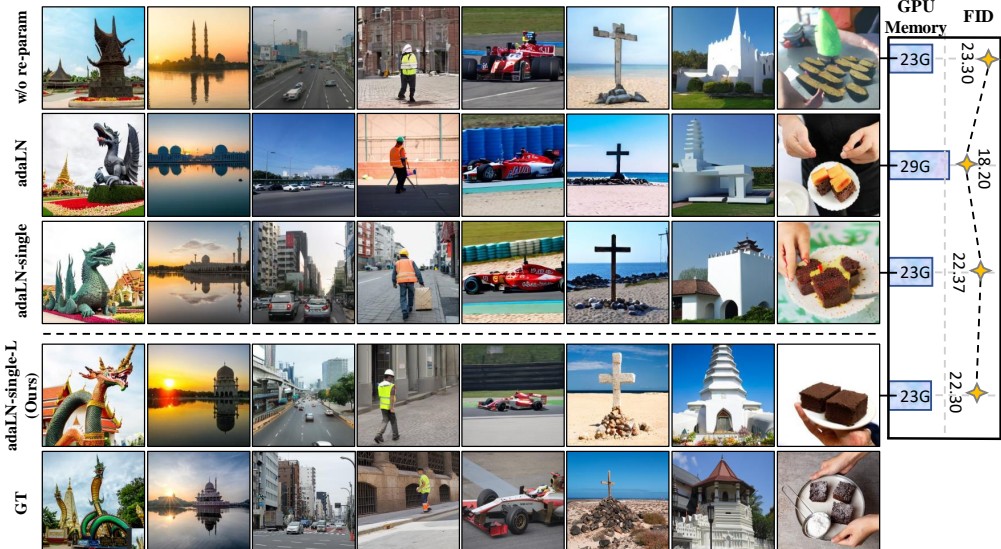

**Figure 6: Left**: Visual comparison of ablation studies are presented. **Right**: Zero-shot FID-2K on SAM, and GPU memory usage. Our method is on par with the "*adaLN*" and saves 21% in GPU memory. Better zoom in 200%.

*single*" achieves a reduction to 23GB, saving 21% in GPU memory consumption. Furthermore, considering the model parameters, the "*adaLN*" method consumes 833M, whereas our approach reduces to a mere 611M, resulting in an impressive 26% reduction. "*adaLN-single-L (Ours)*" results are generated from the model with same setting as "*adaLN-single*", but training for a **L**onger training period of 1500K iterations. Considering memory and parameter efficiency, we incorporate the "*adaLN-single-L*" into our final design.

The visual results clearly indicate that, although the differences in FID scores between the "*adaLN*" and "*adaLN-single*" models are relatively small, a significant discrepancy exists in their visual outcomes. The "*w/o re-param*" model consistently displays distorted target images and lacks crucial details across the entire test set.

# 4    RELATED WORK

We review related works in three aspects: Denoising diffusion probabilistic models (DDPM), Latent Diffusion Model, and Diffusion Transformer. More related works can be found in Appendix A.1. DDPMs (Ho et al., 2020; Sohl-Dickstein et al., 2015) have emerged as highly successful approaches for image generation, which employs an iterative denoising process to transform Gaussian noise into an image. Latent Diffusion Model (Rombach et al., 2022) enhances the traditional DDPMs by employing score-matching on the image latent space and introducing cross-attention-based controlling. Witnessed the success of Transformer architecture on many computer vision tasks, Diffusion Transformer (DiT) (Peebles & Xie, 2023) and its variant (Bao et al., 2023; Zheng et al., 2023) further replace the Convolutional-based U-Net (Ronneberger et al., 2015) backbone with Transformers for increased scalability (Chen et al., 2023).

# 5    CONCLUSION

In this paper, we introduced PIXART-$\alpha$, a Transformer-based text-to-image (T2I) diffusion model, which achieves superior image generation quality while significantly reducing training costs and $CO_2$ emissions. Our three core designs, including the training strategy decomposition, efficient T2I Transformer and high-informative data, contribute to the success of PIXART-$\alpha$. Through extensive experiments, we have demonstrated that PIXART-$\alpha$ achieves near-commercial application standards in image generation quality. With the above designs, PIXART-$\alpha$ provides new insights to the AIGC community and startups, enabling them to build their own high-quality yet low-cost T2I models. We hope that our work inspires further innovation and advancements in this field.

**Acknowledgement.** We would like to express our gratitude to Shuchen Xue for identifying and correcting the FID score in the paper. This research was supported in part by the Research Grants Council of the Hong Kong Special Administrative Region (Grant 16200021).

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

# A APPENDIX

## A.1 RELATED WORK

### A.1.1 DENOISING DIFFUSION PROBABILISTIC MODELS

Diffusion models (Ho et al., 2020; Sohl-Dickstein et al., 2015) and score-based generative models (Song & Ermon, 2019; Song et al., 2021) have emerged as highly successful approaches for image generation, surpassing previous generative models such as GANs (Goodfellow et al., 2014), VAEs (Kingma & Welling, 2013), and Flow (Rezende & Mohamed, 2015). Unlike traditional models that directly map from a Gaussian distribution to the data distribution, diffusion models employ an iterative denoising process to transform Gaussian noise into an image that follows the data distribution. This process can be reversely learned from an untrainable forward process, where a small amount of Gaussian noise is iteratively added to the original image.

### A.1.2 LATENT DIFFUSION MODEL

Latent Diffusion Model (*a.k.a.* Stable diffusion) (Rombach et al., 2022) is a recent advancement in diffusion models. This approach enhances the traditional diffusion model by employing score-matching on the image latent space and introducing cross-attention-based controlling. The results obtained with this approach have been impressive, particularly in tasks involving high-density image generation, such as text-to-image synthesis. This has served as a source of inspiration for numerous subsequent works aimed at improving text-to-image synthesis, including those by Saharia et al. (2022); Balaji et al. (2022); Feng et al. (2023); Xue et al. (2023b); Podell et al. (2023), and others. Additionally, Stable diffusion and its variants have been effectively combined with various low-cost fine-tuning (Hu et al., 2021; Xie et al., 2023) and customization (Zhang et al., 2023; Mou et al., 2023) technologies.

### A.1.3 DIFFUSION TRANSFORMER

Transformer architecture (Vaswani et al., 2017) have achieved great success in language models (Radford et al., 2018; 2019), and many recent works (Dosovitskiy et al., 2020a; He et al., 2022) show it is also a promising architecture on many computer vision tasks like image classification (Touvron et al., 2021; Zhou et al., 2021; Yuan et al., 2021; Han et al., 2021), object detection (Liu et al., 2021; Wang et al., 2021; 2022; Ge et al., 2023; Carion et al., 2020), semantic segmentation (Zheng et al., 2021; Xie et al., 2021; Strudel et al., 2021) and so on (Sun et al., 2020; Li et al., 2022b; Zhao et al., 2021; Liu et al., 2022; He et al., 2022; Li et al., 2022a). The Diffusion Transformer (DiT) (Peebles & Xie, 2023) and its variant (Bao et al., 2023; Zheng et al., 2023) follow the step to further replace the Convolutional-based U-Net (Ronneberger et al., 2015) backbone with Transformers. This architectural choice brings about increased scalability (Chen et al., 2023) compared to U-Net-based diffusion models, allowing for the straightforward expansion of its parameters. In our paper, we leverage DiT as a scalable foundational model and adapt it for text-to-image generation tasks.

## A.2 PIXART-α *vs*. MIDJOURNEY

In Figure 7, we present the images generated using PIXART-α and the current SOTA product-level method Midjourney (Midjourney, 2023) with randomly sampled prompts online. Here, we conceal the annotations of images belonging to which method. Readers are encouraged to make assessments based on the prompts provided. The answers will be disclosed at the end of the appendix.

## A.3 PIXART-α *vs*. PRESTIGIOUS DIFFUSION MODELS

In Figure 8 and 9, we present the comparison results using a test prompt selected by RAPHAEL. The instances depicted here exhibit performance that is on par with, or even surpasses, that of existing powerful generative models.

## A.4 AUTO-LABELING TECHNIQUES

To generate captions with high information density, we leverage state-of-the-art vision-language models LLaVA (Liu et al., 2023). Employing the prompt, "*Describe this image and its style in a very detailed manner*", we have significantly improved the quality of captions. We show the prompt design and process of auto-labeling in Figure 10. More image-text pair samples on the SAM dataset are shown in Figure 11.

## A.5 ADDITIONAL IMPLEMENTATION DETAILS

We include detailed information about all of our PIXART-$\alpha$ models in this section. As shown in Table 4, among the 256×256 phases, our model primarily focuses on the text-to-image alignment stage, with less time on fine-tuning and only 1/8 of that time spent on ImageNet pixel dependency.

**PIXART-$\alpha$ model details.** For the embedding of input timesteps, we employ a 256-dimensional frequency embedding (Dhariwal & Nichol, 2021). This is followed by a two-layer MLP that features a dimensionality matching the transformer's hidden size, coupled with SiLU activations. We adopt the DiT-XL model, which has 28 Transformer blocks in total for better performance, and the patch size of the PatchEmbed layer in ViT (Dosovitskiy et al., 2020b) is 2×.

**Multi-scale training.** Inspired by Podell et al. (2023), we incorporate the multi-scale training strategy into our pipeline. Specifically, We divide the image size into 40 buckets with different aspect ratios, each with varying aspect ratios ranging from 0.25 to 4, mirroring the method used in SDXL. During optimization, a training batch is composed using images from a single bucket, and we alternate the bucket sizes for each training step. In practice, we only apply multi-scale training in the high-aesthetics stage after pretraining the model at a fixed aspect ratio and resolution (*i.e.* 256px). We adopt the positional encoding trick in DiffFit (Xie et al., 2023) since the image resolution and aspect change during different training stages.

**Additional time consumption.** Beside the training time discussed in Table 4, data labeling and VAE training may need additional time. We treat the pre-trained VAE as a ready-made component of a model zoo, the same as pre-trained CLIP/T5-XXL text encoder, and our total training process does not include the training of VAE. However, our attempt to train a VAE resulted in an approximate training duration of 25 hours, utilizing 64 V100 GPUs on the OpenImage dataset. As for auto-labeling, we use LLAVA-7B to generate captions. LLaVA's annotation time on the SAM dataset is approximately 24 hours with 64 V100 GPUs. To ensure a fair comparison, we have temporarily excluded the training time and data quantity of VAE training, T5 training time, and LLaVA auto-labeling time.

**Sampling algorithm.** In this study, we incorporated three sampling algorithms, namely iDDPM (Nichol & Dhariwal, 2021), DPM-Solver (Lu et al., 2022), and SA-Solver (Xue et al., 2023a). We observe these three algorithms perform similarly in terms of semantic control, albeit with minor differences in sampling frequency and color representation. To optimize computational efficiency, we ultimately chose to employ the DPM-Solver with 20 inference steps.

Table 4: We report detailed information about every PIXART-$\alpha$ training stage in our paper. Note that HQ (High Quality) dataset here includes 4M JourneyDB (Pan et al., 2023) and 10M internal data. The count of GPU days excludes the time for VAE feature extraction and T5 text feature extraction, as we offline prepare both features in advance so that they are not part of the training process and contribute no extra time to it.

| Method | Stage | Image Resolution | #Images | Training Steps (K) | Batch Size | Learning Rate | GPU days (V100) |
|---|---|---|---|---|---|---|---|
| PIXART-$\alpha$ | Pixel dependency | 256×256 | 1M ImageNet | 300 | 128×8 | $2\times10^{-5}$ | 88 |
| PIXART-$\alpha$ | Text-Image align | 256×256 | 10M SAM | 150 | 178×64 | $2\times10^{-5}$ | 672 |
| PIXART-$\alpha$ | High aesthetics | 256×256 | 14M HQ | 90 | 178×64 | $2\times10^{-5}$ | 416 |
| PIXART-$\alpha$ | High aesthetics | 512×512 | 14M HQ | 100 | 40×64 | $2\times10^{-5}$ | 320 |
| PIXART-$\alpha$ | High aesthetics | 1024×1024 | 14M HQ | 16 | 12×32 | $2\times10^{-5}$ | 160 |

A.6 HYPER-PARAMETERS ANALYSIS

In Figure 20, we illustrate the variations in the model's metrics under different configurations across various datasets. we first investigate FID for the model and plot FID-vs-CLIP curves in Figure 20a for 10k text-image paed from MSCOCO. The results show a marginal enhancement over SDv1.5. In Figure 20b and 20c, we demonstrate the corresponding T2ICompBench scores across a range of classifier-free guidance (cfg) (Ho & Salimans, 2022) scales. The outcomes reveal a consistent and commendable model performance under these varying scales.

A.7 MORE IMAGES GENERATED BY PIXART-$\alpha$

More visual results generated by PIXART-$\alpha$ are shown in Figure 12, 13, and 14. The samples generated by PIXART-$\alpha$ demonstrate outstanding quality, marked by their exceptional fidelity and precision in faithfully adhering to the given textual descriptions. As depicted in Figure 15, PIXART-$\alpha$ demonstrates the ability to synthesize high-resolution images up to $1024 \times 1024$ pixels and contains rich details, and is capable of generating images with arbitrary aspect ratios, enhancing its versatility for real-world applications. Figure 16 illustrates PIXART-$\alpha$'s remarkable capacity to manipulate image styles through text prompts directly, demonstrating its versatility and creativity.

A.8 DISCCUSION OF FID METRIC FOR EVALUATING IMAGE QUALITY

During our experiments, we observed that the FID (Fréchet Inception Distance) score may not accurately reflect the visual quality of generated images. Recent studies such as SDXL (Podell et al., 2023) and Pick-a-pic (Kirstain et al., 2023) have presented evidence suggesting that the COCO zero-shot FID is negatively correlated with visual aesthetics.

Furthermore, it has been stated by Betzalel et al. (Betzalel et al., 2022) that the feature extraction network used in FID is pretrained on the ImageNet dataset, which exhibits limited overlap with the current text-to-image generation data. Consequently, FID may not be an appropriate metric for evaluating the generative performance of such models, and (Betzalel et al., 2022) recommended employing human evaluators for more suitable assessments.

Thus, we conducted a user study to validate the effectiveness of our method.

A.9 CUSTOMIZED EXTENSION

In text-to-image generation, the ability to customize generated outputs to a specific style or condition is a crucial application. We extend the capabilities of PIXART-$\alpha$ by incorporating two commonly used customization methods: DreamBooth (Ruiz et al., 2022) and ControlNet (Zhang et al., 2023).

**DreamBooth.** DreamBooth can be seamlessly applied to PIXART-$\alpha$ without further modifications. The process entails fine-tuning PIXART-$\alpha$ using a learning rate of 5e-6 for 300 steps, without the incorporation of a class-preservation loss.

As depicted in Figure 17a, given a few images and text prompts, PIXART-$\alpha$ demonstrates the capacity to generate high-fidelity images. These images present natural interactions with the environment under various lighting conditions. Additionally, PIXART-$\alpha$ is also capable of precisely modifying the attribute of a specific object such as color, as shown in 17b. Our appealing visual results demonstrate PIXART-$\alpha$ can generate images of exceptional quality and its strong capability for customized extension.

**ControlNet.** Following the general design of ControlNet (Zhang et al., 2023), we freeze each DiT Block and create a trainable copy, augmenting with two zero linear layers before and after it. The control signal $c$ is obtained by applying the same VAE to the control image and is shared among all blocks. For each block, we process the control signal $c$ by first passing it through the first zero linear layer, adding it to the layer input $x$, and then feeding it into the trainable copy and the second zero linear layer. The processed control signal is then added to the output $y$ of the frozen block, which is obtained from input $x$. We trained the ControlNet on HED (Xie & Tu, 2015) signals using a learning rate of 5e-6 for 20,000 steps.

As depicted in Figure 18, when provided with a reference image and control signals, such as edge maps, we leverage various text prompts to generate a wide range of high-fidelity and diverse images. Our results demonstrate the capacity of PIXART-$\alpha$ to yield personalized extensions of exceptional quality.

## A.10 DISCUSSION ON TRANSFORMER *vs*. U-NET

The Transformer-based network's superiority over convolutional networks has been widely established in various studies, showcasing attributes such as robustness (Zhou et al., 2022; Xie et al., 2021), effective modality fusion (Girdhar et al., 2023), and scalability (Peebles & Xie, 2023). Similarly, the findings on multi-modality fusion are consistent with our observations in this study compared to the CNN-based generator (U-Net). For instance, Table 3 illustrates that our model, PIXART-$\alpha$, significantly outperforms prevalent U-Net generators in terms of compositionality. This advantage is not solely due to the high-quality alignment achieved in the second training stage but also to the multi-head attention-based fusion mechanism, which excels at modeling long dependencies. This mechanism effectively integrates compositional semantic information, guiding the generation of vision latent vectors more efficiently and producing images that closely align with the input texts. These findings underscore the unique advantages of Transformer architectures in effectively fusing multi-modal information.

## A.11 LIMITATIONS & FAILURE CASES

In Figure 19, we highlight the model's failure cases in red text and yellow circle. Our analysis reveals the model's weaknesses in accurately controlling the number of targets and handling specific details, such as features of human hands. Additionally, the model's text generation capability is somewhat weak due to our data's limited number of font and letter-related images. We aim to explore these unresolved issues in the generation field, enhancing the model's abilities in text generation, detail control, and quantity control in the future.

## A.12 UNVEIL THE ANSWER

In Figure 7, we present a comparison between PIXART-$\alpha$ and Midjourney and conceal the correspondence between images and their respective methods, inviting the readers to guess. Finally, in Figure 21, we unveil the answer to this question. It is difficult to distinguish between PIXART-$\alpha$ and Midjourney, which demonstrates PIXART-$\alpha$'s exceptional performance.

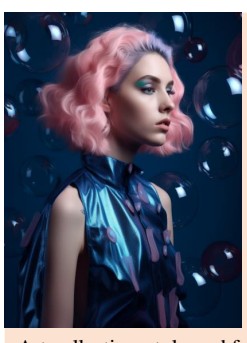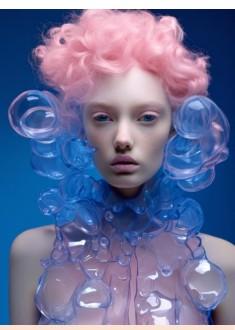

Art collection style and fashion shoot, in the style of made of glass, dark blue and light pink, paul rand, solarpunk, camille vivier, beth didonato hair, barbiecore, hyper-realistic.

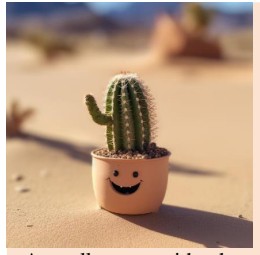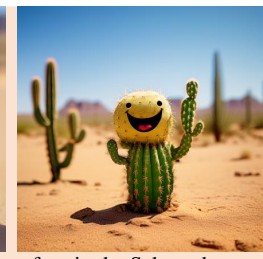

A small cactus with a happy face in the Sahara desert

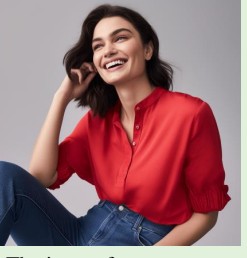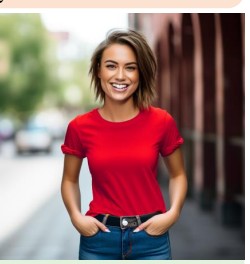

The image features a woman wearing a red shirt with an icon. She appears to be posing for the camera, and her outfit includes a pair of jeans. The woman seems to be in a good mood, as she is smiling. The background of the image is blurry, focusing more on the woman and her attire.

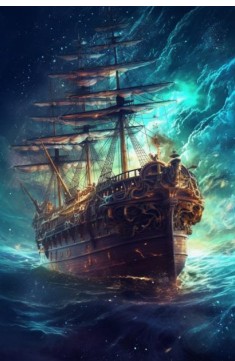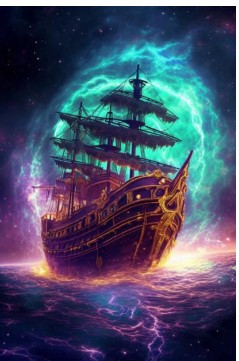

Pirate ship trapped in a cosmic maelstrom nebula, rendered in cosmic beach whirlpool engine, volumetric lighting, spectacular, ambient lights, light pollution, cinematic atmosphere, art nouveau style, illustration art artwork by SenseiJaye, intricate detail.

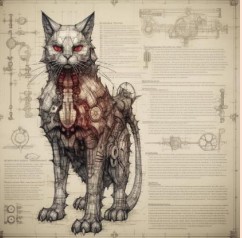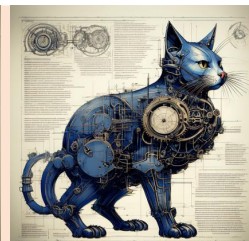

poster of a mechanical cat, technical Schematics viewed from front and side view on light white blueprint paper, illustration drafting style, illustration, typography, conceptual art, dark fantasy steampunk, cinematic, dark fantasy.

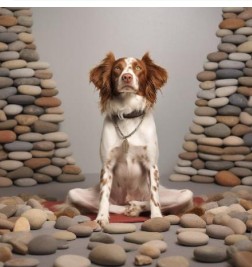

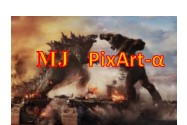

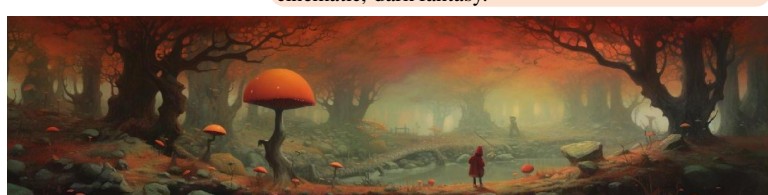

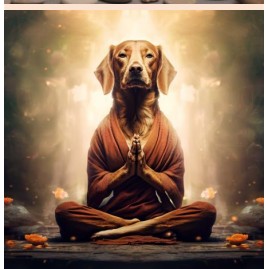

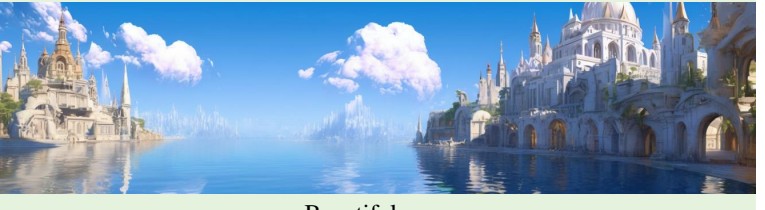

A dog that has been meditating all the time

Beautiful scene

Figure 7: Comparisons with Midjourney. The prompts used here are randomly sampled online. To ensure a fair comparison, we select the first result generated by both models. *We encourage readers to guess which image corresponds to Midjourney and which corresponds to PixArt-α. The answer is revealed at the end of the paper.*

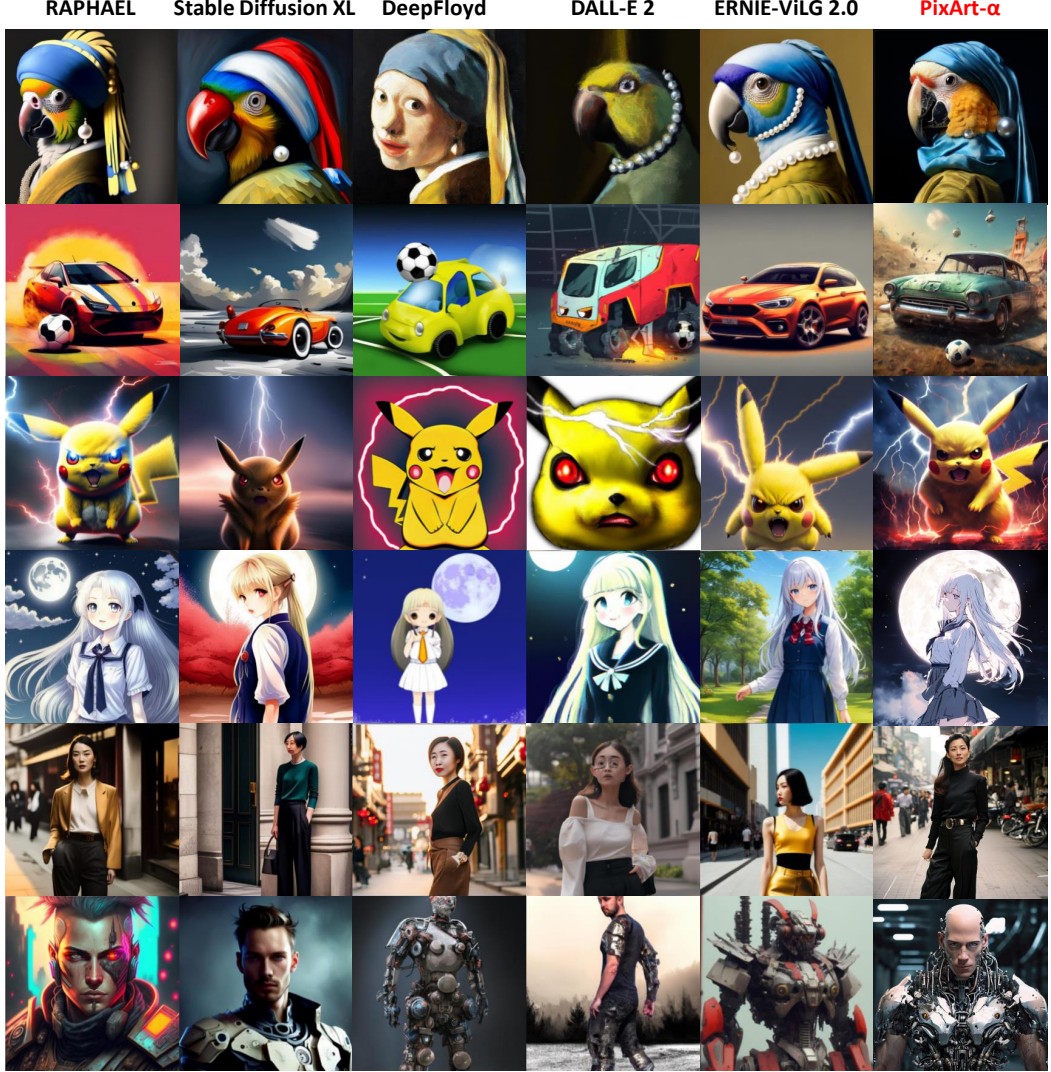

1. A parrot with a *pearl earring*, Vermeer style.

2. A car *playing soccer*, digital art.

3. A Pikachu with an *angry* expression and red eyes, with *lightning* around it, hyper realistic style.

4. Moonlight Maiden, cute girl in school uniform, long *white hair*, standing under the *moon*, celluloid style, *Japanese manga style*.

5. Street shot of a fashionable *Chinese lady* in Shanghai, wearing *black* high-waisted *trousers*.

6. Half *human*, half *robot*, repaired human, human flesh warrior, mech display, man in mech, *cyberpunk*.

Figure 8: **Comparisons** of PIXART-α with recent representative generators, Stable Diffusion XL, DeepFloyd, DALL-E 2, ERNIE-ViLG 2.0, and RAPHAEL. They are given the same prompts as in RAPHAEL(Xue et al., 2023b), where the words that the human artists yearn to preserve within the generated images are highlighted in red. The specific prompts for each row are provided at the bottom of the figure. Better zoom in 200%.

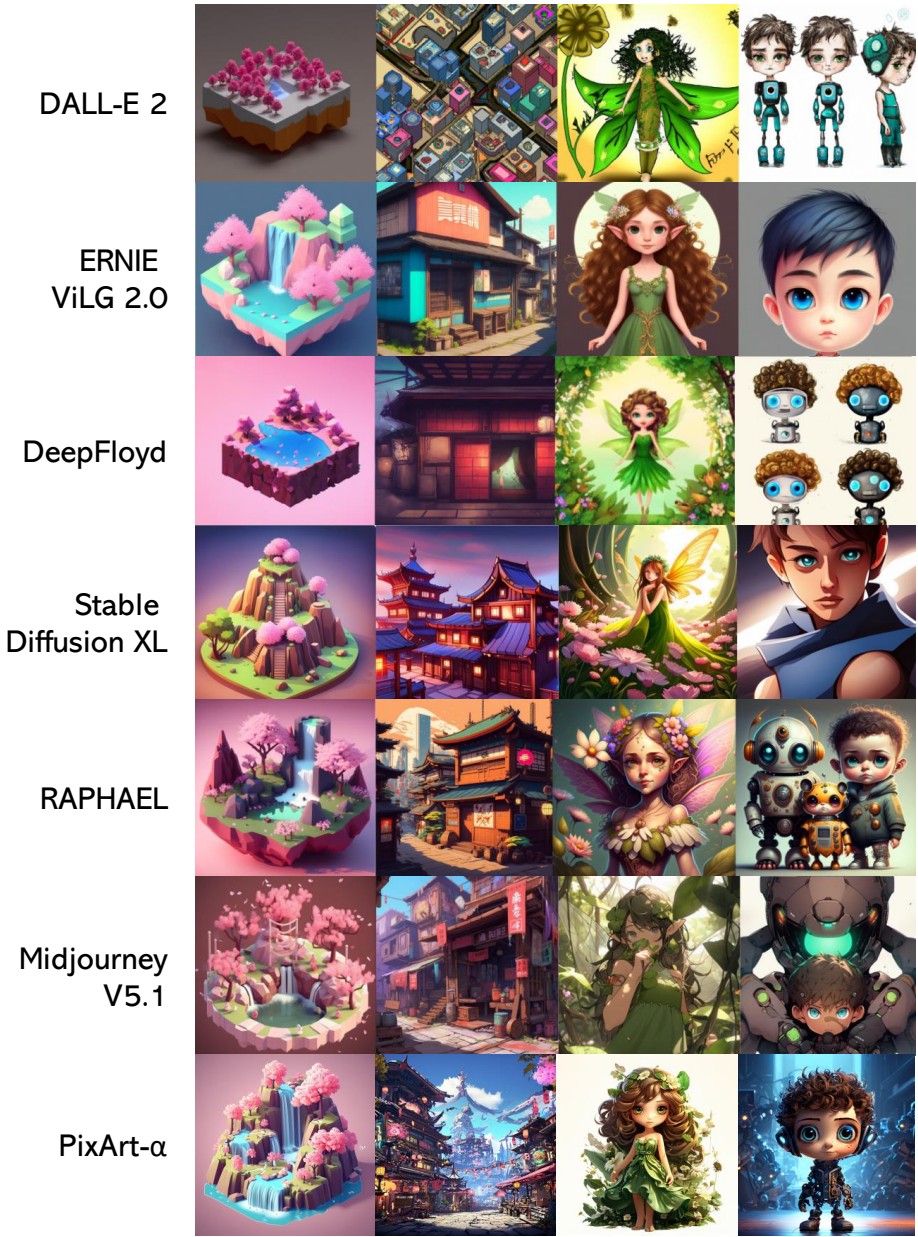

DALL-E 2

ERNIE ViLG 2.0

DeepFloyd

Stable Diffusion XL

RAPHAEL

Midjourney V5.1

PixArt-α

1. A cute little matte low poly isometric *cherry blossom forest island*, *waterfalls*, lighting, soft shadows, trending on Artstation, 3d render, monument valley, fez video game.

2. A shanty version of Tokyo, new rustic style, *bold colors with all colors palette*, video game, genshin, tribe, fantasy, overwatch.

3. Cartoon characters, mini characters, figures, illustrations, flower fairy, green dress, *brown hair, curly long hair, elf-like wings, many flowers and leaves*, natural scenery, *golden eyes*, detailed light and shadow , a high degree of detail.

4. Cartoon characters, mini characters, hand-made, illustrations, *robot kids*, color expressions, boy, *short brown hair*, *curly hair, blue eyes*, technological age, *cyberpunk*, big eyes, cute, mini, detailed light and shadow, high detail.

Figure 9: The prompts (Xue et al., 2023b) for each column are given in the figure. We give the comparisons between DALL-E 2 Midjourney v5.1, Stable Diffusion XL, ERNIE ViLG 2.0, DeepFloyd, and RAPHAEL. They are given the same prompts, where the words that the human artists yearn to preserve within the generated images are highlighted in red. Better zoom in 200%.

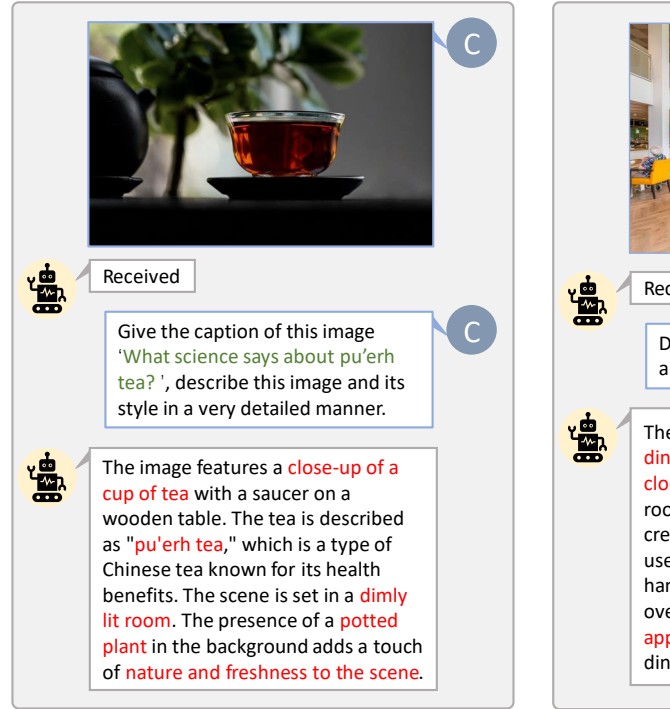
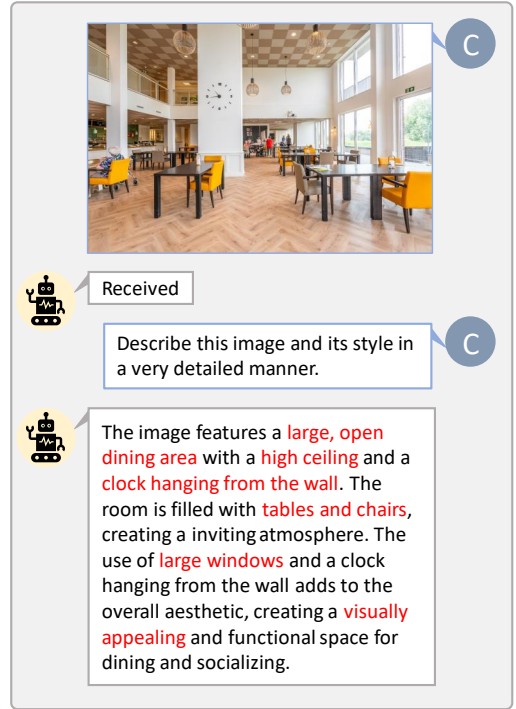

Figure 10: We present auto-labeling with custom prompts for LAION (left) and SAM (right). The words highlighted in green represent the original caption in LAION, while those marked in red indicate the detailed captions labeled by LLaVA.

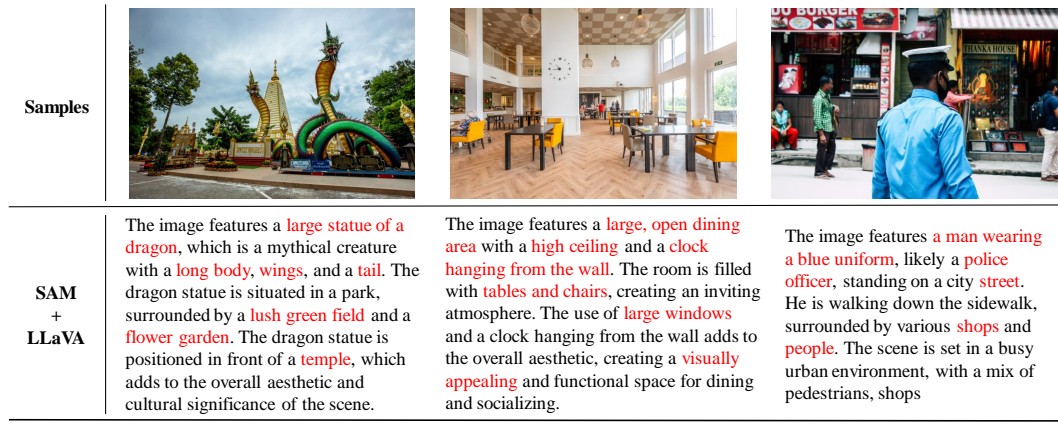

Figure 11: Examples from the SAM dataset using LLaVA-produced labels. The detailed image descriptions in LLaVA captions can aid the model to grasp more concepts per iteration and boost text-image alignment efficiency.

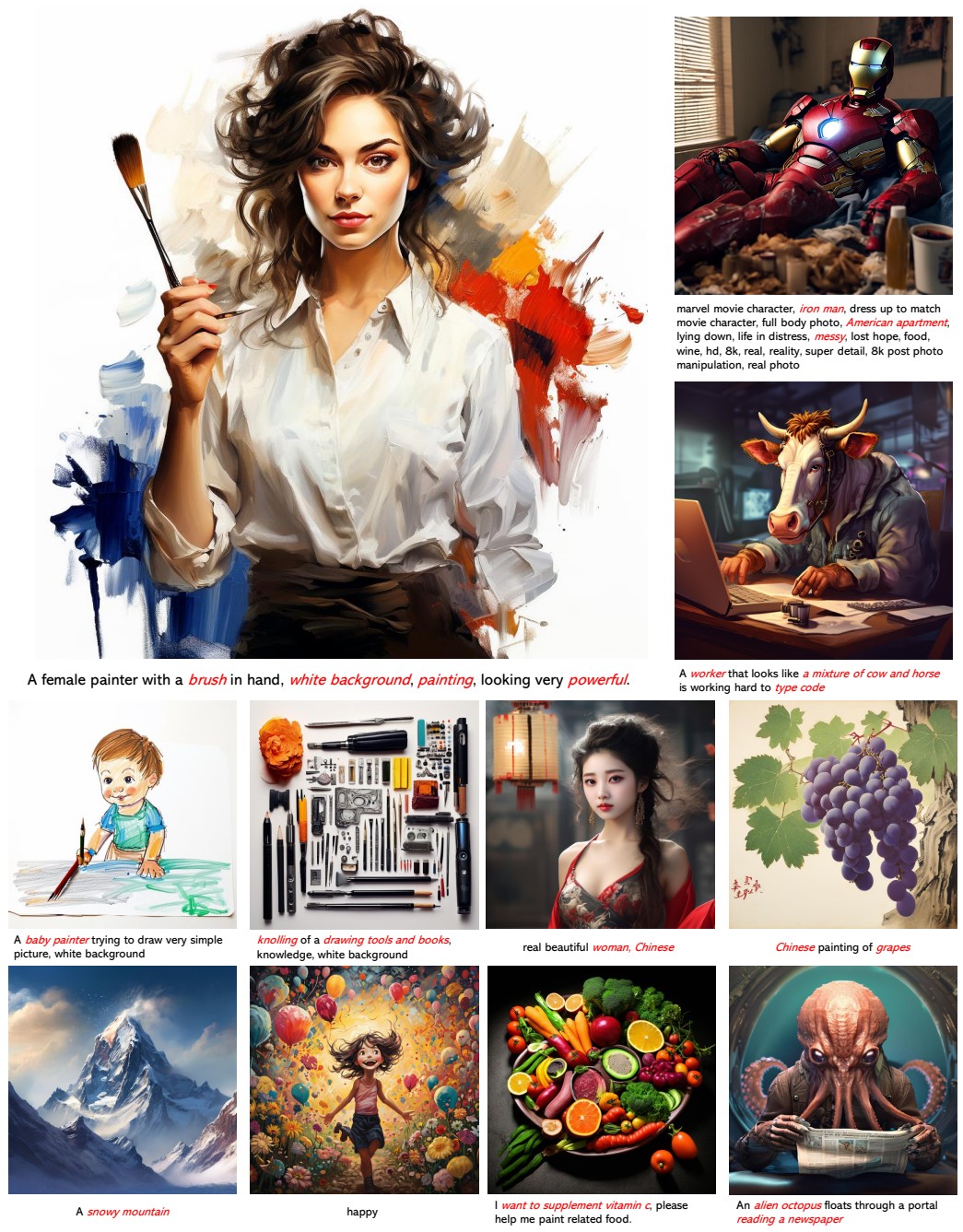

Figure 12: The samples generated by PIXART-α demonstrate outstanding quality, marked by an exceptional level of fidelity and precision in aligning with the given textual descriptions. Better zoom in 200%.

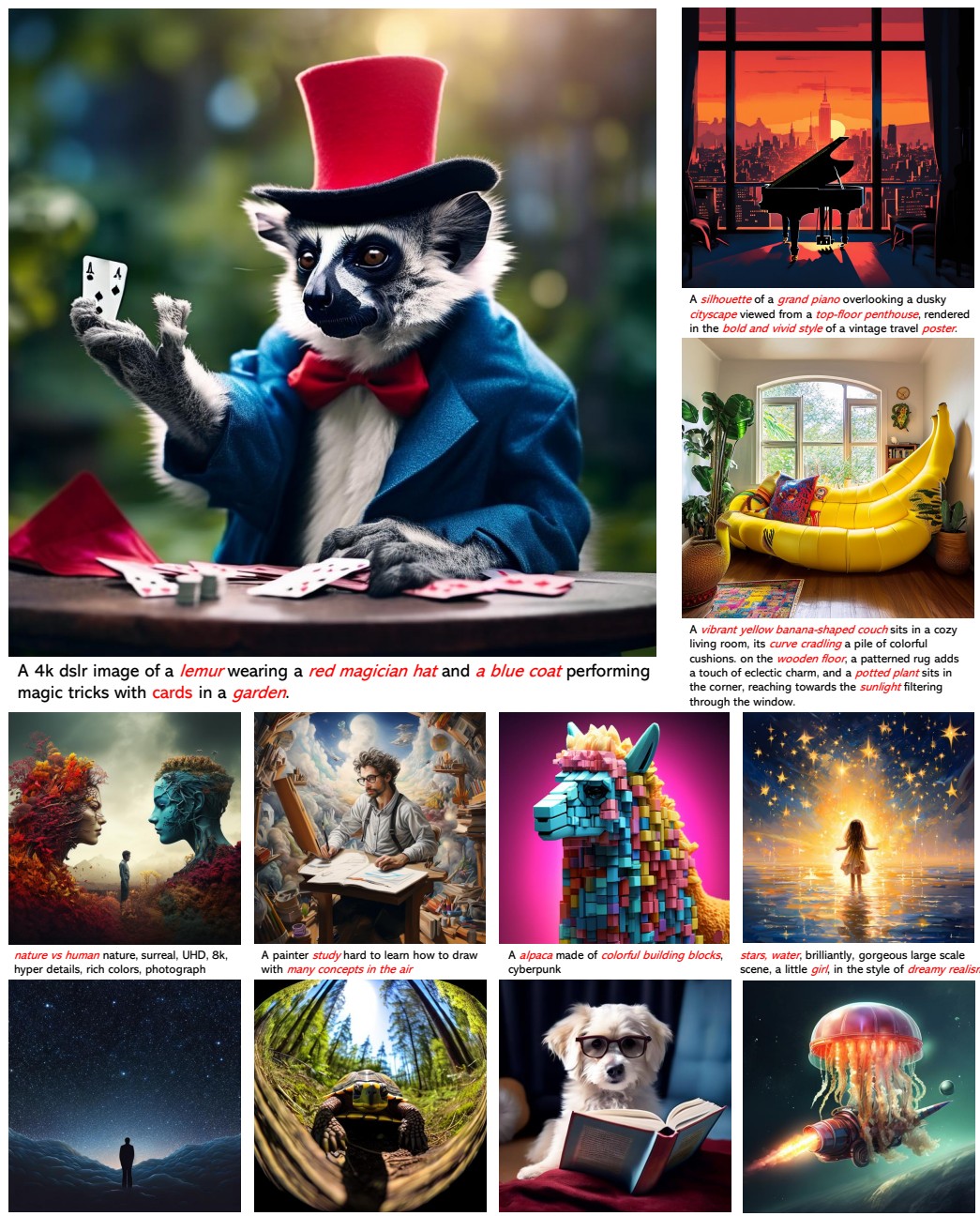

Figure 13: The samples generated by PIXART-$\alpha$ demonstrate outstanding quality, marked by an exceptional level of fidelity and precision in aligning with the given textual descriptions. Better zoom in 200%.

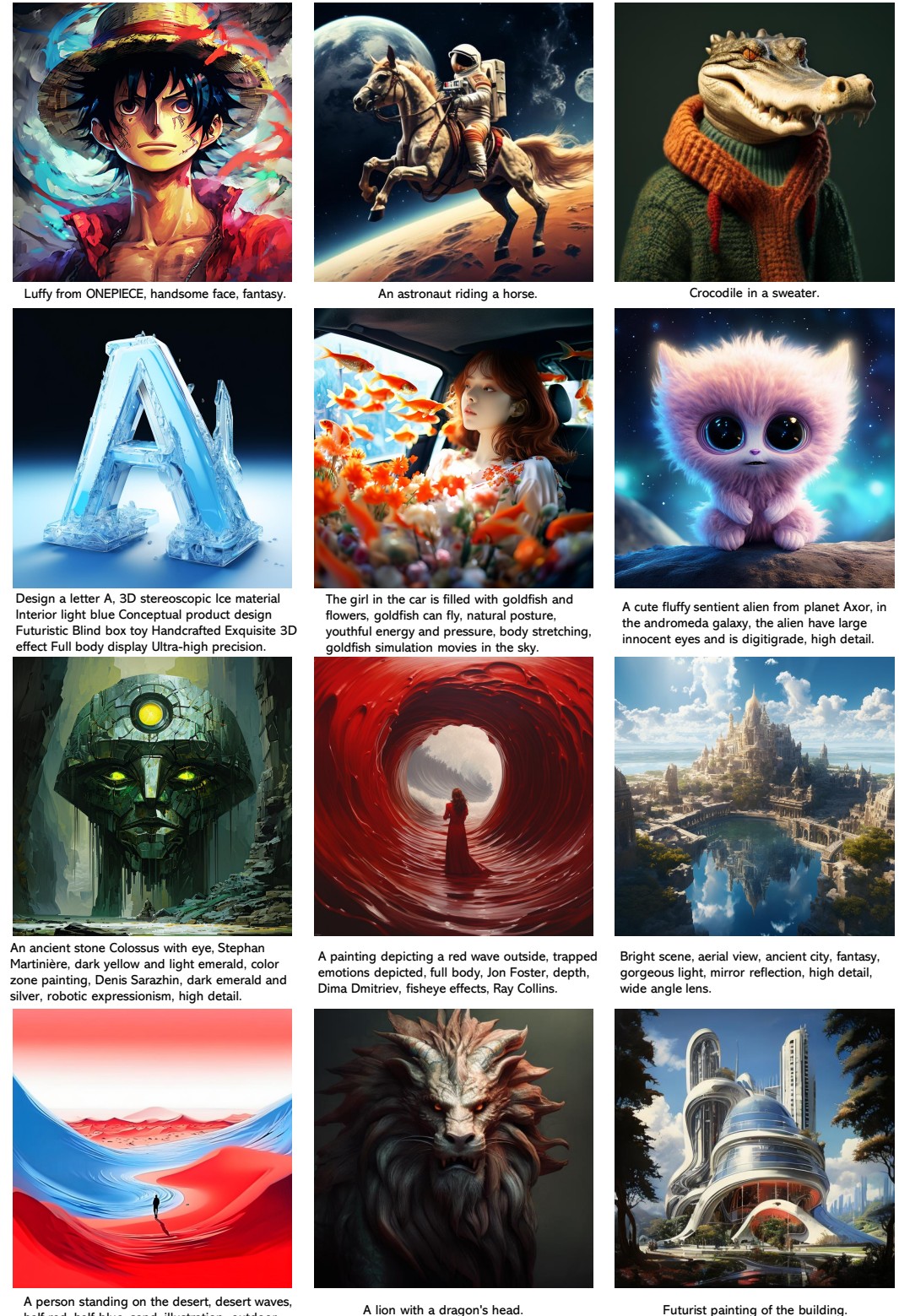

Figure 14: The samples generated by PIXART-$\alpha$ demonstrate outstanding quality, marked by an exceptional level of fidelity and precision in aligning with the given textual descriptions. Better zoom in 200%.

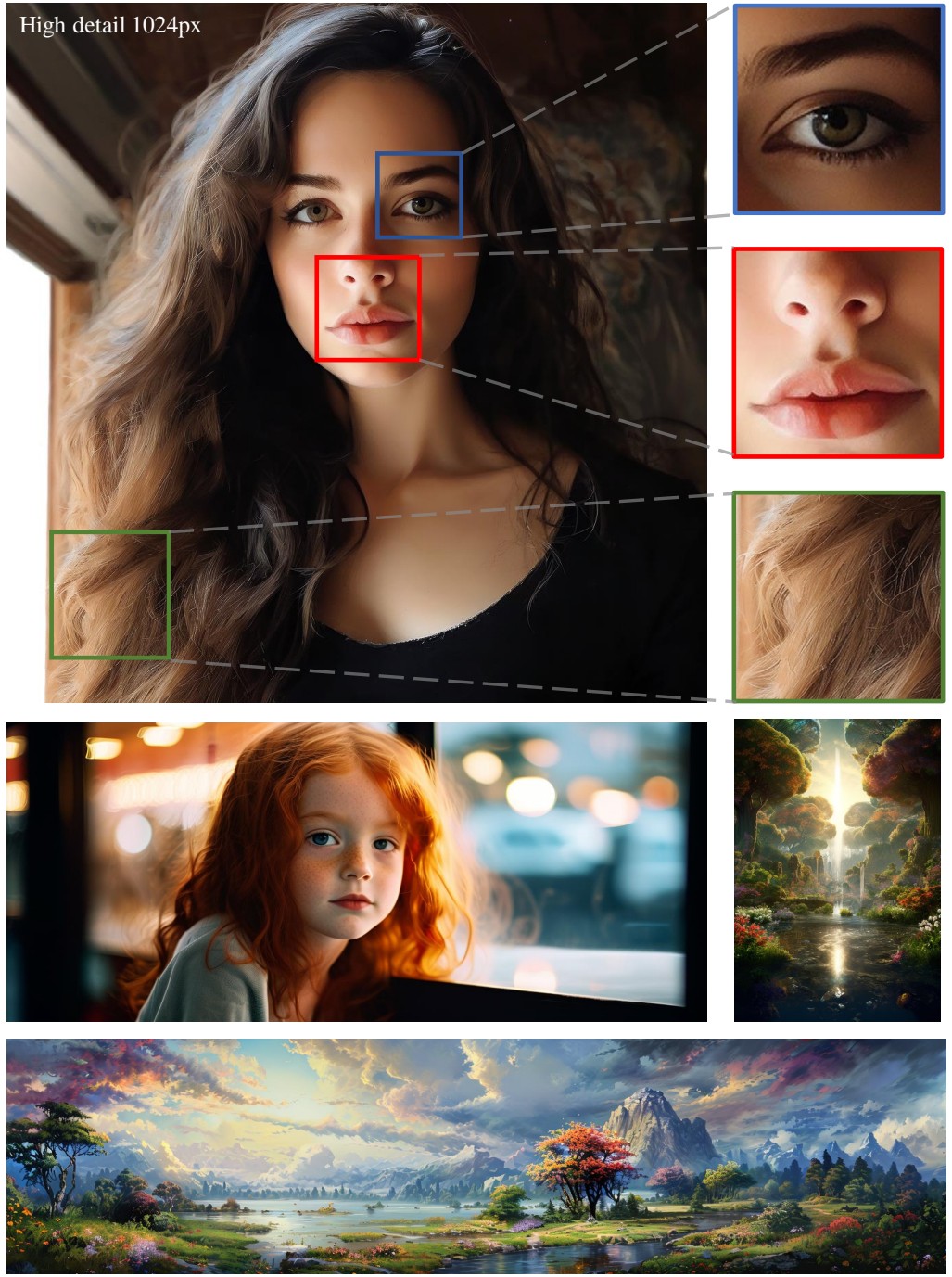

Figure 15: PIXART-$\alpha$ is capable of generating images with resolutions of up to $1024 \times 1024$ while preserving rich, complex details. Additionally, it can generate images with arbitrary aspect ratios, providing flexibility in image generation.

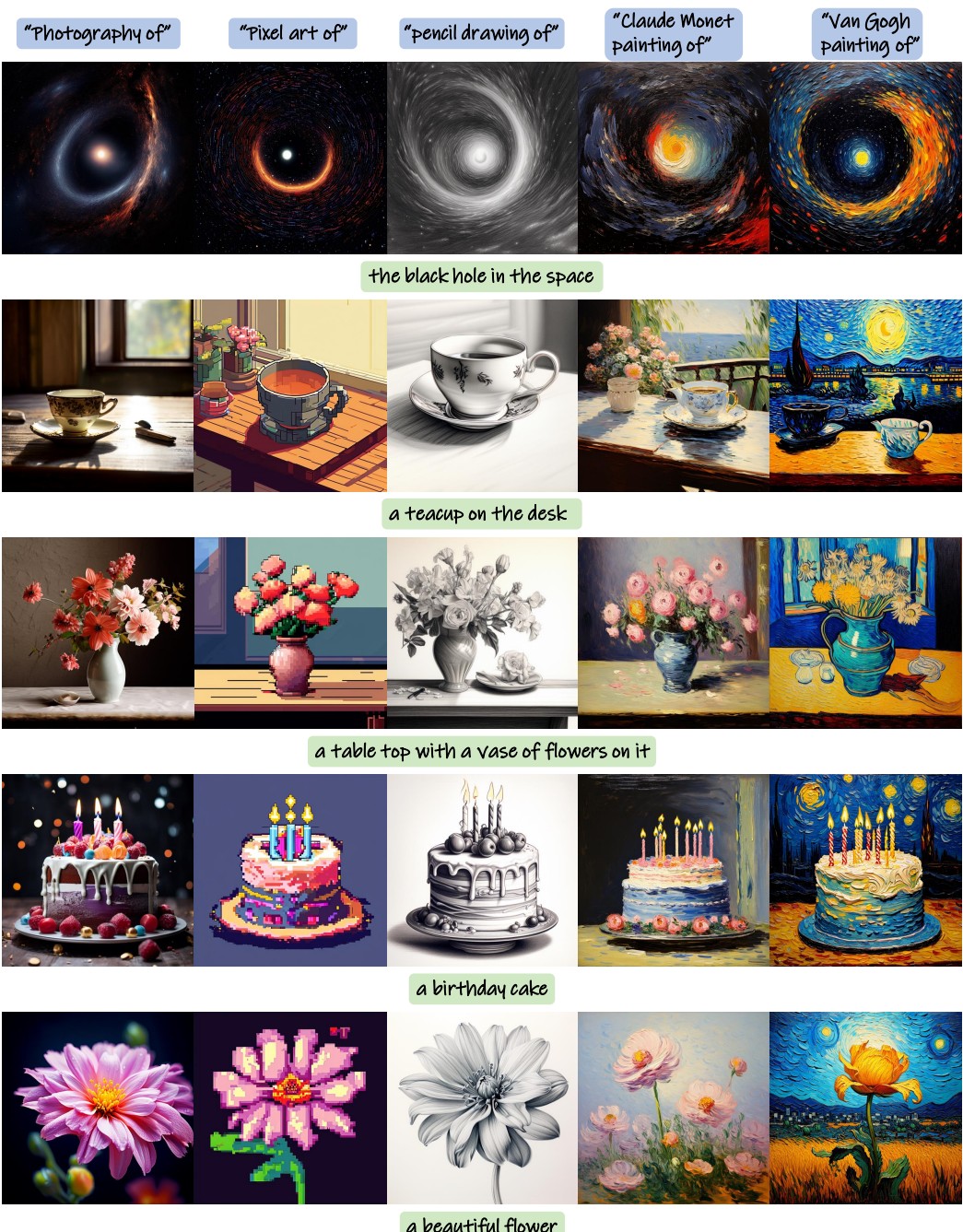

Figure 16: **Prompt mixing:** PIXART-$\alpha$ can directly manipulate the image style with text prompts. In this figure, we generate five outputs using the styles to control the objects . For instance, the second picture of the first sample, located at the left corner of the figure, uses the prompt " Pixel Art of the black hole in the space ". Better zoom in 200%.

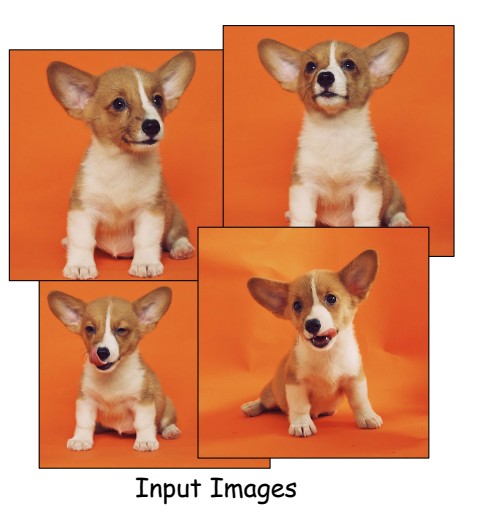

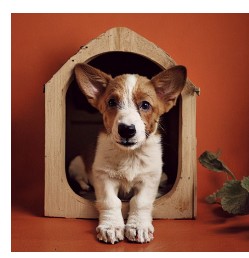

Text prompt:
[V] dog is running

Text prompt:
[V] dog in a doghouse

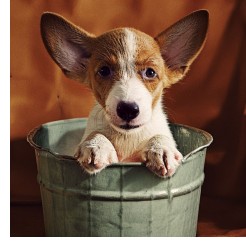

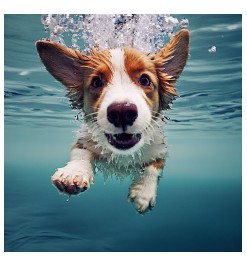

Input Images

Text prompt: A photo of [V] dog

Text prompt:
[V] dog in a bucket

Text prompt:
[V] dog is swimming

(a) Dreambooth + PIXART-$\alpha$ is capable of customized image generation aligned with text prompts.

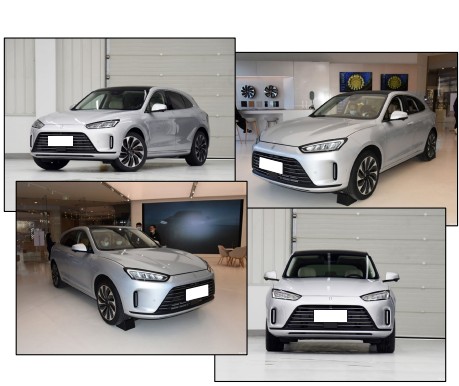

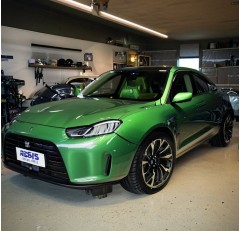

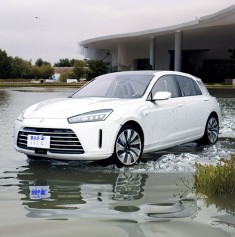

Text prompt:
[green] [V] car in garage

Text prompt:
[white] [V] car over water

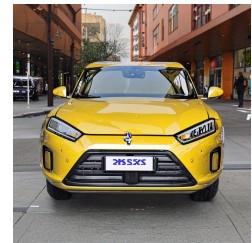

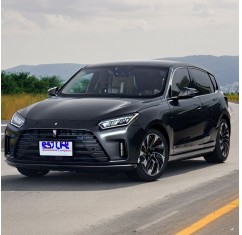

Input Images: 问界M5

Text prompt: A photo of [grey] [V] car

Text prompt:
[yellow] [V] car in street

Text prompt:
[black] [V] car on highway

(b) Dreambooth + PIXART-$\alpha$ is capable of color modification of a specific object such as Wenjie M5.

Figure 17: PIXART-$\alpha$ can be combined with Dreambooth. Given a few images and text prompts, PIXART-$\alpha$ can generate high-fidelity images, that exhibit natural interactions with the environment 17a, precise modification of the object colors 17b, demonstrating that PIXART-$\alpha$ can generate images with exceptional quality, and has a strong capability in customized extension.

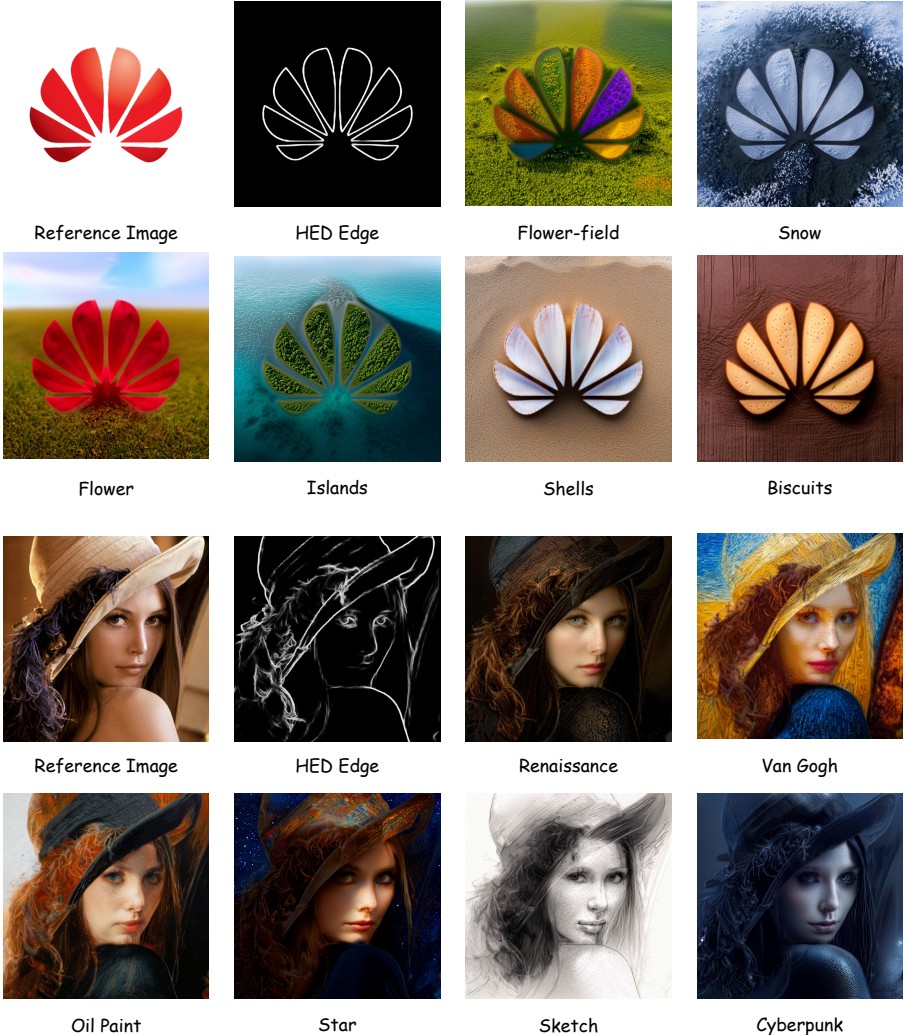

Figure 18: ControlNet customization samples from PIXART-$\alpha$. We use the reference images to generate the corresponding HED edge images and use them as the control signal for PIXART-$\alpha$ ControlNet. Better zoom in 200%.

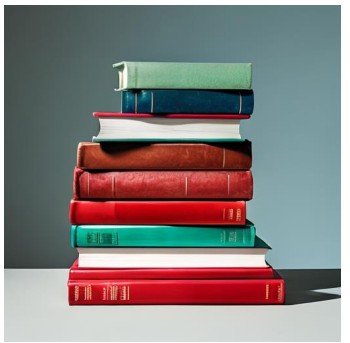 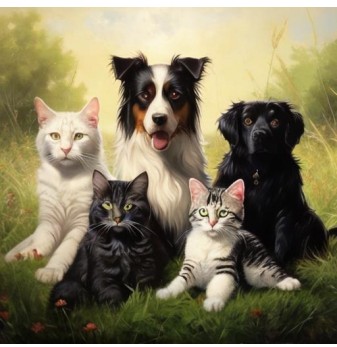 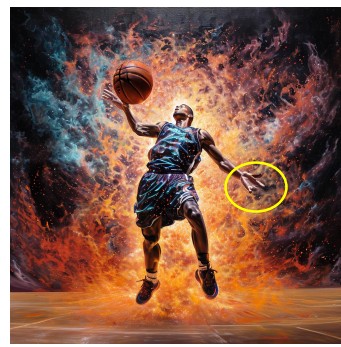

A stack of 3 books. A green book is on the top, sitting on a red book. The red book is in the middle

Three cats and three dogs sitting on the grass

An expressive oil painting of a basketball player dunking, depicted as an explosion of a nebula

Figure 19: Instances where PIXART-$\alpha$ encounters challenges include situations that necessitate precise counting or accurate representation of human limbs. In these cases, the model may face difficulties in providing accurate results.

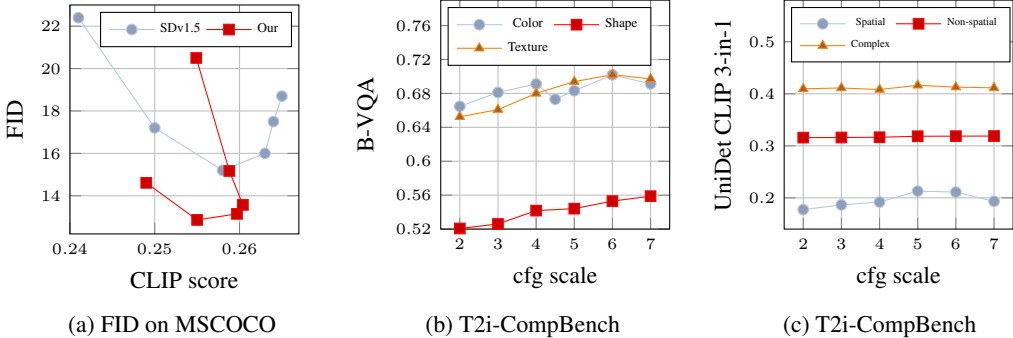

(a) FID on MSCOCO

(b) T2i-CompBench

(c) T2i-CompBench

Figure 20: (a) Plotting FID *vs*. CLIP score for different cfg scales sampled from [1.5, 2.0, 3.0, 4.0, 5.0, 6.0]. PIXART-$\alpha$ shows slight better performance than SDv1.5 on MSCOCO. (b) and (c) demonstrate the ability of PIXART-$\alpha$ to maintain robustness across various cfg scales on the T2I-CompBench.

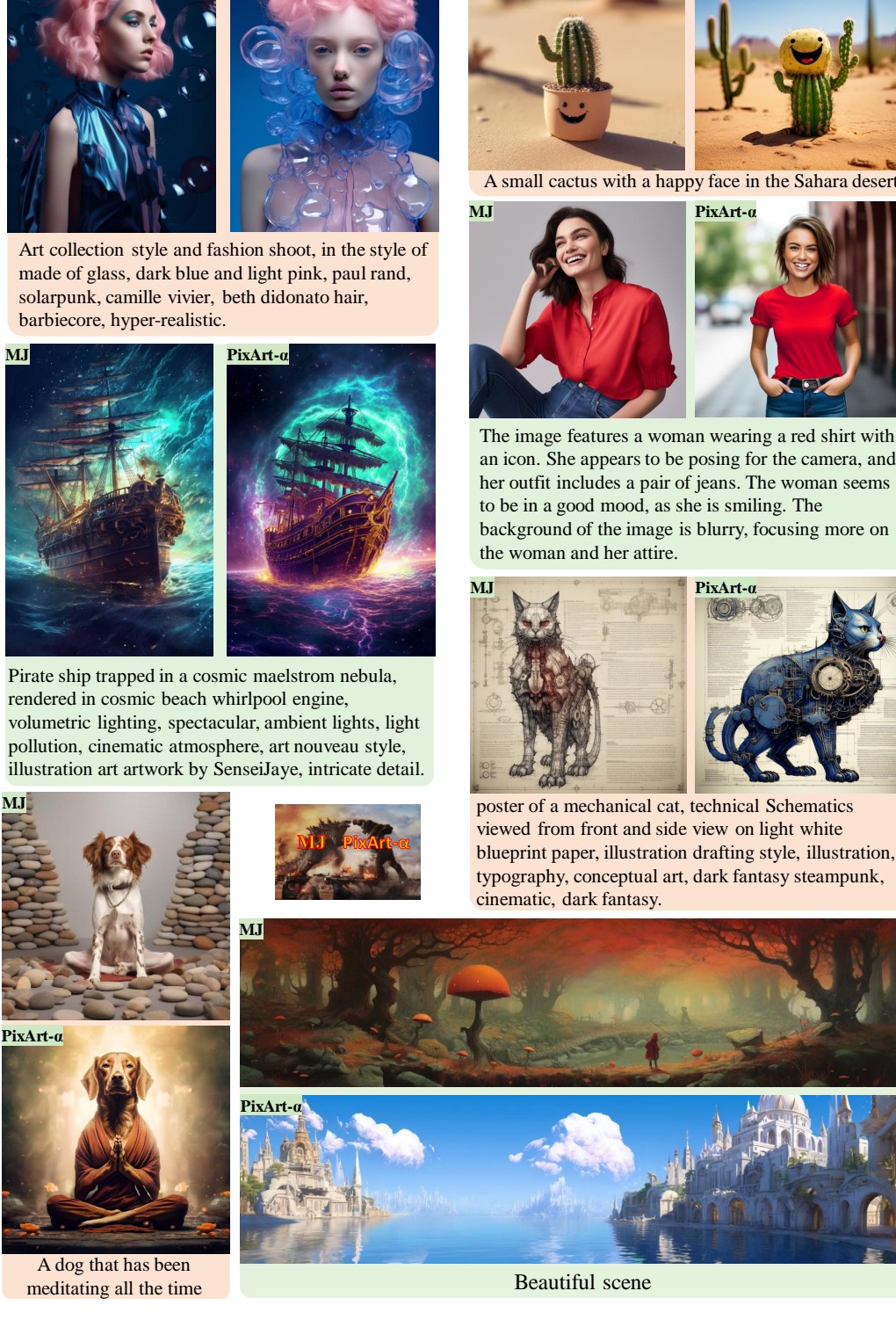

Figure 21: This figure presents the **answers** to the image generation quality assessment as depicted in Appendix A.2. The method utilized for each pair of images is annotated at the top-left corner.

