# OpenReview forum: "PixArt-$\alpha$: Fast Training of Diffusion Transformer for Photorealistic Text-to-Image Synthesis"
_ICLR.cc/2024/Conference — ICLR 2024 spotlight_

### Official Review · Reviewer_7QAx · 2023-10-23

**Soundness:** 2 fair
**Presentation:** 3 good
**Contribution:** 2 fair
**Rating:** 6
**Confidence:** 4

**Summary:**

The authors present a text-to-image model that achieves good quality results while using much smaller networks and datasets than current SOTA models. The key to the improvement seems to be some combination of (1) transformer-based diffusion (2) multi-stage training (5 stages for a single model) (3) careful dataset curation.

**Strengths:**

- Being able to train a high-quality generative model with an ~order of magnitude reduction in resources is a very important contribution, as the cost of training large models limits access to large, well-resourced organizations.

- The results are good, and the evaluations are sufficient. The FID score is a bit high compared to newer models, but I agree with the authors that this is (at best) an imprecise metric for quality.

- The paper is overall well written and easy to read.

**Weaknesses:**

I have found some inconsistencies in reported results:

- The GPU days reported in A.5 contradict the main table; Summing the last column and using their 2.2 scale factor for converting from v100 to a100 days yields 753 A100 days vs. the 675 reported in the main table. It's a relatively small difference (11%) but concerning that the reported numbers don't align.

- Some of the baseline numbers are inconsistent / misleading. For example, the authors report that Imagen consumed 15B images. The Imagen authors report a dataset size of 800M. It seems that 15B, at least roughly, corresponds to the total number of non-unique images seen during training, e.g. (batch size) x (training steps). If we applied this metric to PixArt-alpha (using the table in A.5), we would get a number of 31B - 2x the Imagen results. I did not dig into the other baseline numbers, but these should be checked carefully because they are central to the paper's contributions.

- There are 3 primary improvements (dataset curation, multi-stage training, and unet->transformers), however there are no ablations quantifying how much of an effect each one has on model performance. While the results and final model are good, a lack of ablations limits the research contribution of the paper.

- The authors do not discuss the training of the VAE; since this is an important part of "learning the pixel distribution of natural images", the training time and data quantity of the VAE should be included in the paper.

**Questions:**

- My intuition says that training cost and CO2 emissions should be directly proportional to GPU hours, however these proportions are inconsistent. For example, Fig 2b shows a 14x increase in cost of Imagen over PixArt, while the ratio of reported GPU hours is 10. For GigaGAN, Fig2b shows a 9.4x increase, where as Table 2 shows a 7x increase in GPU hours.

---

> ### Author Response · Authors · 2023-11-16
> **Response to Reviewer 7QAx (1/3)**
>
> Dear Reviewer 7QAx,
>
> We sincerely appreciate your meticulousness and patience in helping us identify the statistical errors in our paper. With your generous help, we believe we can make the article more complete. We will address all issues in the revision, and we hope to receive your approval and forgiveness. We also provide additional clarifications to your comments below.
>
> ---
>
> ### Weakness
>
> **W1: Inconsistency of GPU Days and Data Usage**
>
> We sincerely apologize for the statistical errors in our paper due to the large amount of data information when we wrote this paper. We have fixed all these errors, including the misinterpretation of data referenced from GigaGAN [1] in the revised version. The updates involve GPU days and the training data required by other methods like Imagen [2]. The updated data shows that Pixart uses 5% data volume compared to SDv1.5 and costs less than 2% training time compared to RAPHAEL. Compared to RAPHAEL, our training costs are only 1%, saving approximately \\$3,000,000 (PixArt-α’s  \\$28,400 v.s. RAPHAEL’s \\$3,080,000). All the conclusion **consistently demonstrates the efficiency of PixArt**.
>
> Moreover, Nvidia's tech report [8] illustrates the acceleration ratio between A100 and V100 spec in the first Table below, where the V100->A100 speedup for the Transformer BERT is **5x** v.s. U-Net **2.2x**. We have conservatively estimated the lowest speedup for PixArt in the original paper (we adopted the U-Net speedup ratio). If we consider BERT speedup since PixArt also uses pure Transformer architecture, same as BERT, the training time of PixArt would significantly reduce to **1656 V100 GPU days / 5** = **332 A100 GPU days**.
>
> |         | ResNet-50 v1.5 | U-Net Medical | BERT    |
> | ------- |:--------------:|:-------------:|:-------:|
> | A100    | 4.1 hours      | 3 minutes     | 5 Days  |
> | V100    | 9.7 hours      | 7 minutes     | 24 Days |
> | Speedup | 2.4x           | **2.2x**      | **5x**  |
>
> | Method   | #Images |     GPU days      |
> | -------- |:-------:|:-----------------:|
> | DALL·E   |  250M   |         -         |
> | GLIDE    |  250M   |         -         |
> | LDM      |  400M   |         -         |
> | DALL·E 2 |  650M   |    41,667 A100    |
> | SDv1.5   |  5000M  |    6,250 A100     |
> | GigaGAN  |  2700M  |    4,783 A100     |
> | Imagen   |  860M   |    7,132 A100     |
> | RAPHAEL  | 5000M+  |    60,000 A100    |
> | PIXART-α |   25M   |  753 A100 (2.2x)  |
> | PIXART-α |   25M   | **332 A100** (5x) |

---

> ### Author Response · Authors · 2023-11-16
> **Response to Reviewer 7QAx (2/3)**
>
> **W2: Design of Data, Multi-Stage Training, and UNet->Transformer**
>
> Thanks for this valuable suggestion. Systematically ablation studies of Text-to-Image (T2I) models from model architecture, data curation, and training are computationally expensive and would demand extensive resources, which exceed our current capabilities. However, we still try our best to give some insights and analysis of the above aspects:
>
>
> 1. Data Design:
> * Regarding the selection and re-annotation of LLaVA data, we discussed this in Section 2.4 via the total number of concepts. Additionally, we found that over 90% of the nouns in the LAION concept statistics appear less than 10 times throughout the dataset. Consequently, we believe that the large number of infrequently occurring nouns may be the main reason for prolonging the training time. This is where we attribute the primary cause for rapid training; merely replacing UNet with Transformer wouldn't be enough to accelerate training speed by an order of magnitude.
> * We are continuing to explore the lower limit for the data required in stage 3.
> In stage 3 training, we found that increasing the data volume only marginally improves the model's capabilities, whereas decreasing the data volume has a progressively larger impact.
> We increased the data scale from 14M to 30M but did not observe significant improvements. We also decreased data to 1.4M, 700K, and 140K. The model's performance was relatively stable at 1.4M and 700K, but there was noticeable quality degradation at 140K. Due to time constraints, our current conclusion is that training with 1.4M data points in stage 3 achieves results comparable to 14M. However, volumes below 700K may lead to overfitting, as shown in the below table. The properties of the data should match the VN/DN analysis mentioned in Section 2.4.
>
> | Data Volumn | Iterations | Image size |  FID  | CLIP score |
> |:----------- |:----------:|:----------:|:-----:|:----------:|
> | 30M         |    120K    |    512     | 8.62  |   0.276    |
> | 14M         |    100K    |    512     | 8.61  |   0.275    |
> | 14M         |    15K     |    256     | 16.61 |   0.268    |
> | 1.4M        |    15K     |    256     | 16.96 |   0.266    |
> | 700K        |    15K     |    256     | 17.14 |   0.265    |
> | 140K        |    15K     |    256     | 21.58 |   0.255    |
>
> 2. Multi-stage Training:
> * We tried to skip the first stage and directly train the text-to-image task on SAM. However, due to the significant difference in scale between image and text in cross-attention inputs, and the lack of a corresponding relationship, this usually led to training crashes often represented by NAN. Therefore, we believe that pixel alignment in the first stage is key.
> * The decoupling of stages 2 and 3 is partial because the pseudo-label quality of SAM cannot be guaranteed by current open-source VLMs, leading to illusions, bias, and other issues. Therefore, we only use this data for high-concept-density text-image alignment learning, corresponding to stage 2 pre-training. Fine-tuning with high-aesthetics data can improve the model's aesthetic quality. We have conducted experiments to skip stage 2, and directly train stage 3 based on ImageNet pretrain. Under the same training iterations, we observed that both the visualization results and the text prompt instruction following ability were noticeably inferior compared to using stage 2's SAM pretraining.
> 3. UNet->Transformer:
> * In the original DiT paper, they already thoroughly compared Transformer and UNet architectures on class-condition image generation, validating transformers are easier to scale up and have higher performance upper bound. This is the core reason we chose DiT as the base diffusion model and extended it to the text-to-image model.
> * Moreover, we chose Transformer architecture because it also has several future potentials, including MaskDiT [3], and Window/Sparse/Linear Attention [4~6]. We leave these improvements in the future work.
>
> **W3: VAE Training and Its Data Usage**
>
> Our attempt to train a VAE resulted in an approximate training duration of 25 hours, utilizing 64 V100 GPUs on the OpenImage dataset. Training VAE seems does not consume too much time. We treat the pre-trained VAE as a ready-made component of a model zoo, the same as the pre-trained CLIP/T5-XXL text model, and our total training process did not include the training of VAE and we are also not sure whether other methods e.g. SDXL considered the VAE training time. To ensure a fair comparison, we have temporarily excluded the VAE training time and data quantity. We clarify this point in Appendix A.5 in the revised version of our paper.

---

> > ### Author Response · Authors · 2023-11-16
> > **Response to Reviewer 7QAx (3/3)**
> >
> > ### Questions
> >
> > **Q1: CO2 Emission and Training Cost Misalignment**
> >
> > Sorry to make you confuse about CO2 emissions and training costs. The CO2 consumption here is directly proportional to the **A100 GPU** days because we referred to [7], which only contains A100 CO2 emission Statistics.
> > About the training cost, PixArt is calculated based on the actual training time and price of the **V100 GPU** (refer to https://www.leadergpu.com/). Due to the price differential between the V100 and A100 specs., the ratios of CO2 emissions to training time and cost to training time differ slightly. We will clarify this in the revised version of our manuscript.
> >
> > ---
> >
> > ### Reference
> >
> > [1] Kang M, Zhu J Y, Zhang R, et al. Scaling up gans for text-to-image synthesis[C]//Proceedings of the IEEE/CVF Conference on Computer Vision and Pattern Recognition. 2023: 10124-10134.
> >
> > [2] Saharia C, Chan W, Saxena S, et al. Photorealistic text-to-image diffusion models with deep language understanding[J]. Advances in Neural Information Processing Systems, 2022, 35: 36479-36494.
> >
> > [3] Zheng H, Nie W, Vahdat A, et al. Fast Training of Diffusion Models with Masked Transformers[J]. arXiv preprint arXiv:2306.09305, 2023.
> >
> > [4] Beltagy I, Peters M E, Cohan A. Longformer: The long-document transformer[J]. arXiv preprint arXiv:2004.05150, 2020.
> >
> > [5] Child R, Gray S, Radford A, et al. Generating long sequences with sparse transformers[J]. arXiv preprint arXiv:1904.10509, 2019.
> >
> > [6] Choromanski K, Likhosherstov V, Dohan D, et al. Rethinking attention with performers[J]. arXiv preprint arXiv:2009.14794, 2020.
> >
> > [7] Anne-Laure Ligozat Alexandra Sasha Luccioni, Sylvain Viguier. Estimating the carbon footprint of bloom, a 176b parameter language model. In arXiv preprint arXiv:2211.02001, 2022.
> >
> > [8] NVIDIA. Getting immediate speedups with a100 and tf32, 2023. URL https://developer. nvidia.com/blog/getting-immediate-speedups-with-a100-tf32.

---

> ### Author Response · Authors · 2023-11-21
> **Response to Reviewer 7QAx**
>
> Dear Reviewer 7QAx,
>
> Thank you for your detailed review and the valuable feedback. We have carefully addressed each of your concerns and provided clarifications and experiment in our previous response. We would like to kindly request your response to the provided explanations and revisions.
>
> We appreciate your thorough evaluation of our work and look forward to hearing from you and addressing any further questions or concerns you may have.
>
> Thank you for your continued engagement and support.

---

> > ### Comment · Reviewer_7QAx · 2023-11-23
> >
> > The authors have satisfied many of my concerns. I've raised my score.

---

> > > ### Author Response · Authors · 2023-11-23
> > > **Author Response**
> > >
> > > Dear 7QAx,
> > >
> > > We sincerely thank the reviewer again for the detailed discussions and the kind support of this work. Your constructive feedback and criticisms will help us greatly towards improving this work.

---

### Official Review · Reviewer_Bg3n · 2023-10-30

**Soundness:** 4 excellent
**Presentation:** 4 excellent
**Contribution:** 4 excellent
**Rating:** 8
**Confidence:** 4

**Summary:**

The paper proposes a new text-to-image synthesis model, called PixArt-$\alpha$. The proposed approach introduces several improvements to conventional models on the side of the training process and architecture. In particular, the paper proposes a training strategy decomposition, in which 1) the learning of dependencies between pixels; 2) text-image alignment; 3) high-resolution synthesis, are all modelled in different subsequent phases. On the architecture side, the paper builds on top of Diffusion Transformer pre-trained on class-conditional ImageNet, proposing several tricks to adopt it to text-to-image synthesis with minimal overhead. In addition, the authors re-visit the training datasets used to train text-to-image models. By applying LLaVA to label existing datasets, much more dense labelling of images in terms of nouns/image is achieved.

In effect, the proposed approach achieves state-of-the-art quality of synthesis but significantly reduces the training time and the needed amount of training data. The method is shown to be on par or better than models like Imagen, SDXL, while using only 10% of the training time and 10% of the training samples.

**Strengths:**

Overall, this is a very actual paper with very strong experimental result, big significance and potential impact.

- $\underline{\text{Contribution}}$. The proposed method achieves state-of-the-art results with resources that are an order of magnitude lower than of current mainstream methods. Given that biggest T2I are very expensive to train, this is a big step forward for the community to democratize T2I models for broader audiences, to decrease costs and the carbon footprint required for training.
- $\underline{\text{Insights}}$. The paper introduces valuable insights for the community, particularly on the side of dataset design. The paper demonstrates that the design of previous datasets like LAION suffers from deficient descriptions and infrequent diverse vocabulary usage, which led previous models for many additional epochs needed until convergence. The provided VN/DN analysis can influence the next generation of image-text datasets.
- $\underline{\text{Results}}$. The paper demonstrates strong experimental results. In particular, visual results are impressive, beautiful, and have good alignment to conditioning text. The model is shown to outperform big models like DALLE-2 or SDXL by human perception. Overall, the quality of results clearly matches the bar or ICLR.
- $\underline{\text{Applications}}$. It is demonstrated that the proposed method supports applications that are generally expected from big T2I models, like personalization (DreamBooth, ControlNet etc).
- $\underline{\text{Presentation}}$. The paper is very well written, the ideas are very easy to follow. Explanations are high-level and clearly deliver lessons for the community. More technical discussions are presented in supplementary.

**Weaknesses:**

I do not see major issues that would preclude publication, but I would still ask a couple of questions:

- How expensive was the dataset collection? I expect running a big image-text model like LLaVA on millions of images required a lot of time, GPU ressources, and memory. Should this non-trivial step be in some way included to the analysis of costs and co2 emissions?
- Although using a model like LLava clearly improves VN/DN, it probably introduces new biases to the distribution of text prompts? Does this affect the scope of text prompts for which PixArt-$\alpha$ works well or poorly?

**Questions:**

What are the plans of the authors regarding open-sourcing?
Is it expected to be released 1) complete training code; 2) collected dataset; 3) "internal" dataset?

---

> ### Author Response · Authors · 2023-11-16
> **Response to Reviewer Bg3n**
>
> Dear Reviewer Bg3n,
>
> Thank you very much for your appreciation and high praise of our article. We will take your suggestions into account to make the article even better. We address your comments below.
>
> ---
>
> ### Questions
>
> **Q1: Cost and Resource Requirements for LLaVA auto-captioning.**
>
> We use LLAVA-7B to generate captions. LLaVA's annotation time on the SAM dataset is approximately 24 hours with 64 V100; it is negligible compared to training time. Following your suggestion, we clarify this point in Appendix A.5 in the revised version of our paper.
>
> **Q2: Potential Bias Introduced by LLaVA**
>
> LLaVA may introduce certain illusions and biases, but it does not harm the learning of concepts and text-image alignment. Therefore, we only use LLaVA-caption for text-image alignment pre-training. In stage 3, we finetune our internal data with GT text prompts (users given) to quickly adapt to user preference. Our main contribution is a proof of concept that using VLM models like LLaVA for captioning is reasonable and practical. As VLM rapidly evolves, some new VLMs, e.g. GPT-4V, can produce better auto-labeling captions and significantly contribute to T2I pre-training.
>
> **Q3: Open Source Planning**
>
> We have released training/inference code, checkpoints, and demos, which have had a certain impact on the AIGC community.

---

> > ### Comment · Reviewer_Bg3n · 2023-11-21
> > **Thanks for getting back**
> >
> > I thank the authors for their answers. I remain positive in my assessment and have no further questions to the authors.

---

> > > ### Author Response · Authors · 2023-11-23
> > > **Author Response**
> > >
> > > Dear Bg3n,
> > >
> > > We sincerely thank the reviewer for the constructive feedback and support.

---

### Official Review · Reviewer_GFA3 · 2023-11-01

**Soundness:** 4 excellent
**Presentation:** 3 good
**Contribution:** 3 good
**Rating:** 6
**Confidence:** 4

**Summary:**

PIXART-α is a novel Text-to-Image (T2I) model capable of generating photorealistic images from text descriptions while maintaining low training costs. The model's approach involves decomposing the T2I task into three distinct stages, which include pixel dependency learning, text-image alignment learning, and high-resolution and aesthetic image generation. This is achieved through the modification of the diffusion transformer architecture, incorporating cross-attention layers, simplifying adaptive normalization layers, and utilizing re-parameterization techniques for an efficient T2I Transformer. Additionally, the model leverages high-informative data from a vision-language model (LLaVA) for generating quality image captions, drawing from the SAM dataset and fine-tuning using JourneyDB and an internal dataset.

**Strengths:**

The PIXART-α model is particularly cost-effective to train, making it an attractive option for researchers and organizations with limited computational resources, as it minimizes the financial and hardware requirements associated with training a state-of-the-art T2I model.

It employs a straightforward and simplified approach to achieve its text-to-image synthesis, ensuring that the model's architecture and training process are accessible and comprehensible to a broader range of researchers and practitioners.

Notably, PIXART-α boasts fewer tunable parameters, which not only contributes to its cost-effectiveness but also makes it more manageable and less prone to overfitting or complex hyperparameter tuning, streamlining the implementation and optimization process.

**Weaknesses:**

Given its smaller dataset, there are concerns about the generalizability and compositional capabilities of the model for a broader range of concepts.

A more extensive test for out-of-domain generalizability would add valuable insights into the model's adaptability and effectiveness in diverse scenarios.

**Questions:**

What led to the decision of utilizing ImageNet data as the pretraining source instead of a pretrained VQGAN, and how did this choice impact the model's performance and capabilities?

The selection of the Diffusion Transformer as the basis for the model's architecture over the more conventional approach of the Latent Diffusion Model (Unet) is mentioned in Appendix A10, but can a more comprehensive analysis, including ablation studies, be provided to thoroughly compare the strengths and weaknesses of these architectural choices?

The model's performance raises the important question of determining the minimum number of images required for training while still ensuring generalizability and maintaining the compositional properties of Text-to-Image models. Can the authors shed some light on what is the minimum set of images required to train such a model from scratch. What should be the properties of such a dataset?

---

> ### Author Response · Authors · 2023-11-16
> **Response to Reviewer GFA3 (1/2)**
>
> Dear Reviewer GFA3,
>
> Thank you for appreciating our approach. We address your comments below.
>
> ---
>
> ### Weakness
>
> **W1: Concerns About OoD Generalization with Smaller Dataset.**
>
> We believe there are no generalizability issues in PixArt. As detailed in Section 2.4, although our total training samples are fewer than LAION-5B, each sample contains much more information, leading to better overall information volume and density. Specifically, as shown in **Table 1**, the total number of valid distinct nouns (VN) in SAM and Internal data is 152K + 23K = **175K**, which is comparable with LAION's **210K**. Moreover, we empirically found that for some OoD prompts, such as surreal image generation, like the prompt in Figure 1 where *"a cactus with a happy face"*, SDXL fails to produce satisfactory images while PixArt's generated image is good.
>
> In summary, we can conclude that our dataset scale is big enough, and PixArt does not suffer from generalizability issues.
>
> ---
> ### Questions
>
> **Q1: Using ImageNet-Pretrained DiT Over Pretrained VQGAN**
>
> We apologize for the confusion regarding the use of ImageNet. We explain our choice in two aspects:
>
> 1. Image encoder/decoder choice: We chose VAE over VQGAN [1], following current open-source popular methods like SDXL [2], SD [3], and DiT [4]. The choice between VAE and VQGAN is not the primary focus of our paper, and using VQGAN as an alternative is also feasible.
> 2. Genrative model choice: We chose the Diffusion Transformer instead of the Auto-regressive Transformer in VQGAN, mainly because Denoising Diffusion Probabilistic Models (DDPMs) are the most mainstream and mature data modeling method for image generation, currently most of the advanced image generative models are based on DDPMs. Meanwhile, auto-regressive Transformer for image generation is also promising since some recent papers, e.g. CM3Leon (meta), also adopted this solution.
>
> **Q2: Comprehensive Ablation of UNet vs Transformer Architectures**
>
> In the original DiT paper, they already thoroughly compared Transformer and UNet architectures on class-condition image generation, validating transformers are easier to scale up and have higher performance upper bound. This is the initial reason we chose DiT as the base diffusion model.
>
> However, systematically assessing the impact of different model architectures in the Text-to-Image (T2I) domain requires considering several factors, such as the model size, data volume, and computational requirements. Therefore, exploring the effectiveness of various architectures in the T2I field would demand extensive resources, which exceed our current capabilities. Our primary motivation for this project is to accelerate training T2I model under limited resources. The core contributions of our paper are how to use data more efficiently and how to smartly decouple the training process to speed up training T2I models ultimately.
>
>
> **Q3: Insights about Minimum Scale and Properties of the Dataset**
>
> That's an excellent question. We are continuing to explore the lower limit for the data required in stage 3. Below is our observation.
>
> In stage 3 training, we found that increasing the data volume only marginally improves the model's capabilities, whereas decreasing the data volume has a progressively larger impact.
>
> Specifically, we increased the data scale from 14M to 30M but did not observe significant improvements. We also decreased data to 1.4M, 700K, and 140K. The model's performance was relatively stable at 1.4M and 700K, but there was noticeable quality degradation at 140K. Due to time constraints, our current conclusion is that training with 1.4M data points in stage 3 achieves results comparable to 14M. However, volumes below 700K may lead to overfitting, as shown in the below table. The properties of the data should match the VN/DN analysis mentioned in Section 2.4.
>
> | Data Volumn | Iterations | Image size | FID   | CLIP score |
> |:-----------:|:----------:|:----------:|:-----:|:----------:|
> | 30M         | 120K       | 512        | 8.62  | 0.276      |
> | 14M         | 100K       | 512        | 8.61  | 0.275      |
> | 14M         | 15K        | 256        | 16.61 | 0.268      |
> | 1.4M        | 15K        | 256        | 16.96 | 0.266      |
> | 700K        | 15K        | 256        | 17.14 | 0.265      |
> | 140K        | 15K        | 256        | 21.58 | 0.255      |
>
> ---

---

> > ### Author Response · Authors · 2023-11-16
> > **Response to Reviewer GFA3 (2/2)**
> >
> > ### Reference
> >
> > [1] Esser P, Rombach R, Ommer B. Taming transformers for high-resolution image synthesis[C]//Proceedings of the IEEE/CVF conference on computer vision and pattern recognition. 2021: 12873-12883.
> >
> > [2] Dustin Podell, Zion English, Kyle Lacey, Andreas Blattmann, Tim Dockhorn, Jonas M ̈ uller, Joe Penna, and Robin Rombach. Sdxl: Improving latent diffusion models for high-resolution image synthesis. In arXiv, 2023.
> >
> > [3] Robin Rombach, Andreas Blattmann, Dominik Lorenz, Patrick Esser, and Bj ̈ orn Ommer. Highresolution image synthesis with latent diffusion models. In CVPR, 2022.
> >
> > [4] William Peebles and Saining Xie. Scalable diffusion models with transformers. In ICCV, 2023.

---

> ### Author Response · Authors · 2023-11-21
> **Response to Reviewer GFA3**
>
> Dear Reviewer GFA3,
>
> Thank you for your detailed review and the valuable feedback. We have carefully addressed each of your concerns and provided clarifications and experiment in our previous response. We would like to kindly request your response to the provided explanations and revisions.
>
> We appreciate your thorough evaluation of our work and look forward to hearing from you and addressing any further questions or concerns you may have.
>
> Thank you for your continued engagement and support.

---

### Official Review · Reviewer_VQLf · 2023-11-01

**Soundness:** 4 excellent
**Presentation:** 3 good
**Contribution:** 4 excellent
**Rating:** 8
**Confidence:** 4

**Summary:**

The paper introduces several recipes to accelerate the training of text-to-image foundation models. These include
- Use pretrained DiT
- Use Flan-T5 XXL
- Use LLaVA captions.
- AdaLN / AdaLN single architectures to reduce model size.
Overall, these methods allow training of a reasonable quality model in 10% of the resources used than Stable Diffusion, which makes it more possible to democratize the training recipes in text-to-image foundation models.

**Strengths:**

Originality: the empirical evaluation of synthetic captions in text to image generation is not systematically studied until DALL-E 3, and the new AdaLN architecture might be useful.
Clarify: the paper is quite clear about most details about the training, which makes reproducibly much more likely.
Significance: the paper mostly explores valid heuristics for training text-to-image foundation models quickly, some conclusions can be helpful in the community: 1) DiT architecture instead of UNet, 2) the use of synthetic captions, 3) the use of SAM dataset.

**Weaknesses:**

The paper mostly is a combination of multiple ideas that exist in the literature, so "novelty" in the traditional sense is somewhat limited.

**Questions:**

1. The SAM dataset blurs human faces in their training, won't this cause problem in generation cases where generating a face (not closed up) is needed?
2. How does the model generate images with more extreme aspect ratios?
3. The DiT architecture has a fixed patch size. As resolution becomes higher, so will the number of tokens be higher. Will this cause a bottleneck in training and inference (such as 1k resolution)?
4. DALL-E 3 technical report mentions the pitfall of "overfitting" to automated captions, is this the case in PixArt model? If not, how is this mitigated?
5. Since the dataset size is smaller, does it have trouble producing named entities, such as celebrities?
6. How critical is training DiT on ImageNet needed? While being able to start with an existing model is good it also limits the possibilities to explore different architectures.
7. The CLIP score of the CLIP-FID curve of Pixart seems worse than SD 1.5. Is there any reason for that?

---

> ### Author Response · Authors · 2023-11-16
> **Response to Reviewer VQLf (1/2)**
>
> Dear Review VQLf,
>
> Thank you for appreciating our approach. We will address your comments below.
>
> ---
>
> ### Weakness
>
> **W1: Combination of Multiple Ideas.**
>
> We understand the 'traditional novelty' mentioned in the review. We will discuss more about the novelty here.
> 1. **Data**. Our paper's primary contribution to the community lies in the auto-labeling approach we have developed, a topic that has yet to see much discussion thus far. Coupled with the analysis of concept density, as detailed in Section 2.4, we provide a fresh perspective on assessing the suitability of a dataset for diffusion training.
> 2. **Training process**. Decoupling the entire training process into multiple stages is another innovative finding in our work. This approach significantly accelerates the training process, unlike previous methods [1,2], which decoupled the training process into different resolutions. While these earlier methods enhance the model's forward pass speed, we contribute to faster learning.
> 3. **Model structure**. we design an adaptation technique to validate the feasibility of transfer from class-condition to Text-to-Image (T2I) field and make it outperform existing methods.
>
> In summary, our paper presents novel findings, insights, and designs concerning data, the training process, and model structure. We believe our approach could inspire the community in future research endeavors.
>
> ---
>
> ### Questions
> **Q1: Concerns about Blurry Faces in SAM Dataset.**
>
> The blurry faces in the SAM dataset indeed pose challenges in pretraining models to generate clear faces. However, we primarily utilize SAM for Text-Image Alignment learning, where high image quality is not crucial. In later high-quality fine-tuning phases, we have ample data with clear facial images, allowing for rapid rectification of such issues.
>
> **Q2: How to Generate Images with Extreme Aspect Ratios?**
>
> Drawing inspiration from SDXL [1], we pre-define various aspect ratios ranging from 1:4 to 4:1 and train batches of images with the same aspect ratio. We also add aspect ratios (AR) as one additional condition to the model, allowing the model to be aware of different ARs. During inference, users can specify a specific AR and a corresponding noise map with the same AR will be initiated and input into the model. Technically, PixArt can easily generate images with arbitrary aspect ratios. However, most of the training data's aspect ratio ranges from 1:4 to 4:1, so we cannot guarantee the quality of the generated images with an extreme aspect ratio.
>
>
> **Q3: Efficiency of Training/Inference When Image Resolution Becomes Higher.**
>
> Increasing image sizes in transformers will negatively impact training/inference efficiency. Therefore, in our work, most training (stages 1,2, and part of stage 3) was conducted at 256px size and only finetuning several steps at 512px/1024px size. Training on 32G V100 GPUs, we do not observe significant bottleneck within 1K resolution. However, larger resolutions, such as 2K, might pose challenges due to memory constraints. Several potential solutions can be used to improve the Transformer's efficiency, including MaskDiT [3], and Window/Sparse/Linear Attention [4~6]. We leave it for future work.

---

> ### Author Response · Authors · 2023-11-16
> **Response to Reviewer VQLf (2/2)**
>
> **Q4: How PixArt Mitigate the Oitfall of "Overfitting" to Automated Captions Compared with DALL-E 3?**
>
> DALL-E 3 and PixArt follow different training processes. DALL-E 3 uses a mixture of ground truth (GT) and pseudo captions during training. In contrast, PixArt initially uses  LLaVA-generated pseudo captions for pretraining (stage 2), then finetuning with GT text-image pairs (stage 3). Considering the captioning quality of LLaVA and similar VLM models are still under enhancement, the automated captions are mainly used for image-text alignment learning. Moreover, in stage 3, the model also learns text-image alignment; the only difference is that the captions of stage 3 are provided by real users (Section 2.4). After stage 3 finetuning, the model effectively aligns with users' preferences/habits, enabling PixArt to avoid overfitting from pseudo captions.
>
> **Q5: Generation Named Entities (e.g. celebrities), Given Smaller Dataset Size**
>
> PixArt has a certain ability to generate images of well-known people such as Donald Trump and Elon Musk, but it can't guarantee accuracy for individuals not present in the dataset. We test and observe that even when the Stable Diffusion is trained on a 5B volume dataset, its ability to generate celebrity images remains limited. To ease this problem, we can add related datasets for training. e.g. CelebA [7], which encompasses an array of celebrity faces.
>
> **Q6: Benefits of Pretraining DiT on ImageNet and Restriction on Model Exploration**
>
> Pretraining on ImageNet does not limit model exploration. Instead, it enhances training stability.
> 1. Directly training a random-initialized DiT on the SAM dataset led to frequent NaN issues because the models need to learn image denoising and text-image alignment simultaneously.
> 2. We have validated the feasibility and efficiency of extending the class-conditioned model to the text-conditioned model; this part is orthogonal to the network architecture.
>
> So it is conceptually simple to explore different architectures to pretrain on ImageNet and finetune on the SAM dataset without restrictions.
>
> **Q7: Reasons for Lower CLIP Scores Compared to SD1.5**
>
> PixArt and SD1.5 exhibit comparable CLIP scores.
> 1. Considering both FID and CLIP scores, the comparison between PixArt and SD1.5 is evenly matched, with PixArt demonstrating a higher CLIP score at the lowest FID point, as shown in Fig. 20.
> 2. The datasets used by SD and PixArt differ: the SD utilizes the LAION-5B dataset and filters the images with CLIP scores < 0.28 [8]. Training with the data already having high CLIP scores might present an advantage.
> 3. We tested the ground truth image-text pair CLIP score on MSCOCO-3W, which is only **0.257**, and we found that both SD1.5 and PixArt exceed this score. Therefore, we believe this metric can not effectively reflect the true performance of the advanced T2I models.
>
> ---
>
> ### Reference
> [1] Dustin Podell, Zion English, Kyle Lacey, Andreas Blattmann, Tim Dockhorn, Jonas M ̈ uller, Joe Penna, and Robin Rombach. Sdxl: Improving latent diffusion models for high-resolution image synthesis. In arXiv, 2023.
>
> [2] DeepFloyd. Deepfloyd, 2023. URL https://www.deepfloyd.ai/.
>
> [3] Zheng H, Nie W, Vahdat A, et al. Fast Training of Diffusion Models with Masked Transformers[J]. arXiv preprint arXiv:2306.09305, 2023.
>
> [4] Beltagy I, Peters M E, Cohan A. Longformer: The long-document transformer[J]. arXiv preprint arXiv:2004.05150, 2020.
>
> [5] Child R, Gray S, Radford A, et al. Generating long sequences with sparse transformers[J]. arXiv preprint arXiv:1904.10509, 2019.
>
> [6] Choromanski K, Likhosherstov V, Dohan D, et al. Rethinking attention with performers[J]. arXiv preprint arXiv:2009.14794, 2020.
>
> [7] Liu Z, Luo P, Wang X, et al. Large-scale celebfaces attributes (celeba) dataset[J]. Retrieved August, 2018, 15(2018): 11.
>
> [8] Schuhmann C, Beaumont R, Vencu R, et al. Laion-5b: An open large-scale dataset for training next generation image-text models[J]. Advances in Neural Information Processing Systems, 2022, 35: 25278-25294.

---

### Author Response · Authors · 2023-11-22
**To ALL**

We sincerely appreciate all reviewers for their time and efforts in reviewing our paper. We are pleased to note that reviewers have acknowledged the following contributions to our work:
* **Resource Efficiency and Accessibility ([VQLf, GFA3, Bg3n, 7QAx]):** The proposed PIXART-α model significantly reduces the resources required for training high-quality Text-to-Image (T2I) generative models. This advancement makes state-of-the-art T2I technology more accessible to a broader audience, democratizing the field and reducing both financial and environmental costs.
* **Innovative Dataset Design and Strong Experimental Results ([GFA3, Bg3n, 7QAx]):** The paper offers valuable insights into dataset design. We provide a fresh perspective on assessing the suitability of a dataset for diffusion training in terms of description quality and vocabulary diversity. The new approach not only accelerates model convergence but also achieves impressive visual results and good text alignment, outperforming large models like DALL-E 2 and Stable Diffusion in user study.
* **Simplified Approach with Broad Application Potential ([VQLf, Bg3n]):** PIXART-α employs a straightforward architecture with fewer tunable parameters, making it more manageable and less prone to overfitting. The model supports a wide range of applications expected from large T2I models, such as personalization and content control, demonstrating its versatility and potential impact in various domains.

These contributions highlight the work's significance in advancing T2I model training efficiency, offering new perspectives on dataset optimization, and broadening the practical applications of generative models.

---

We also thank all reviewers for their insightful and constructive suggestions, which helped further improve our paper. In addition to the pointwise responses below, we summarize the major revisions in the rebuttal according to the reviewers' suggestions.
* **Cost and Resource Requirements for LLaVA auto-captioning and VAE training ([Bg3n, 7QAx]):** For a better understanding of the overall training efficiency, we included detailed auto-labeling and VAE training duration in the revised version.

* **Extended Experiments ([GFA3, 7QAx]):** We continue to explore the lower limit for the data required in stage 3. Below is our observation.
In stage 3 training, we found that increasing the data volume only marginally improves the model's capabilities, whereas decreasing the data volume has a progressively larger impact.
Specifically, we increased the data scale from 14M to 30M but did not observe significant improvements. We also decreased data to 1.4M, 700K, and 140K. The model's performance was relatively stable at 1.4M and 700K, but there was noticeable quality degradation at 140K. Due to time constraints, our current conclusion is that training with 1.4M data points in stage 3 achieves results comparable to 14M. However, volumes below 700K may lead to overfitting, as shown in the below table. The properties of the data should match the VN/DN analysis mentioned in Section 2.4.

| Data Volumn | Iterations | Image size | FID   | CLIP score |
|:----------- |:---------- |:---------- |:----- |:---------- |
| 30M         | 120K       | 512        | 8.62  | 0.276      |
| 14M         | 100K       | 512        | 8.61  | 0.275      |
| 14M         | 15K        | 256        | 16.61 | 0.268      |
| 1.4M        | 15K        | 256        | 16.96 | 0.266      |
| 700K        | 15K        | 256        | 17.14 | 0.265      |
| 140K        | 15K        | 256        | 21.58 | 0.255      |

* **Updating Manuscript ([7QAx]):** As recommended, we have corrected all errors and misunderstandings in the revised version of our paper. (contents in blue)


We hope our pointwise responses below can clarify reviewers' confusion and alleviate all concerns. We thank all reviewers' efforts and time again. **With the deadline approaching, we warmly welcome and look forward to any additional discussions that may arise post-deadline.**

Best,

Authors

---

### Meta-Review · Area_Chair_pm9s · 2023-12-08

**Metareview:**

This paper proposes a novel architecture and training recipe for latent diffusion models. Reducing the training cost by a factor 10 for similar performance. The author rebuttal addressed most of the points raised by the reviewers. One reviewer raised their score in the light of the rebuttal. All reviewers support acceptance of the paper.

**Justification For Why Not Higher Score:**

This a "systems" paper that combines several existing techniques to obtain the speedup in training text-to-image diffusions models. Therefore I feel the technical novelty does not warrant a full oral.

**Justification For Why Not Lower Score:**

The paper is potentially useful for a large audience of researchers working on text-to-image diffusion models. To me this potential impact warrants highlighting it as a spotlight.

---

### Decision · Program_Chairs · 2024-01-16

Accept (spotlight)